



# Thinning of the Quelccaya Ice Cap over the last thirty years

C. D. Chadwell[1], D. R. Hardy[2], C. Braun[3], H. H. Brecher[4], and L. G. Thompson[4]

[1]Marine Physical Lab, Scripps Institution of Oceanography, Univ. of California San Diego, La Jolla, CA 92093-0205, USA.
[2]Department of Geosciences, Univ. of Massachusetts, Amherst, MA 01003-9297, USA.
[3]Geography and Regional Planning / Environmental Science, Westfield State Univ., Westfield, MA 01086, USA.
[4]Byrd Polar and Climate Research Center, The Ohio State University, Columbus, OH 43210, USA.

*Correspondence to:* C. D. Chadwell (cchadwell@ucsd.edu)

**Abstract.** Direct measurements of the decadal response of Tropical glaciers to environmental changes are difficult to acquire within their accumulation zones. In 2013, we used dual-frequency kinematic GPS to re-measure the surface elevations at 46 sites, from the margin to across the summit of the Quelccaya Ice Cap, first measured in 1983 using terrestrial surveying methods. In 2015, six additional sites on the western margin, first observed in 1978, were remeasured. Over the past 30 years, the ice cap summit has thinned by $4.41 \pm 0.23$ m ($2\sigma$), with a maximum ice loss at one site near the margin of $63.4 \pm 0.34$ m ($2\sigma$) over 37 years. Using geophysical methods that located the sub-glacial bedrock, we estimate the unit-volume of ice in 1983 along a profile from the 1983 margin to the summit and then the change in volume from 1983 to 2013 by differencing the surface elevations. Over the past 30 years, $21.2 \pm 0.3\%$ ($2\sigma$) of the ice unit-volume has been lost suggesting an average annual mass balance rate of $-0.5 \pm 0.1$ m w.e a$^{-1}$ ($2\sigma$). Increasing air temperature at high elevations of the Andes is likely a major driver of the observed changes. Specifically, within the ablation zone, thinning is likely caused by a 1-2 m w.e. a$^{-1}$ increase in melting and sublimation above steady-state. Within the accumulation zone, analysis of annual, dry-season summit pits suggests that surface lowering may be caused by both a slight decrease in net snow accumulation and an increase in firnification rate, though this interpretation yet lacks statistical significance. The role of ice flux changes since 1983/4 remains unconstrained, awaiting updated measurements of ice surface velocities across the ice cap.

## 1 Introduction

The Quelccaya Ice Cap (QIC, 13°56′S, 70°50′W, Fig. 1) is located ∼5300-5670 meters above sea level (masl) in the southeastern Andes of Peru and is the Earth's largest tropical glacier. Field studies began in 1974 with shallow pits dug at the summit that showed annual snow accumulation of ∼3 m, with apparent annual concentrations of microparticles and oxygen isotopes (Thompson and Dansgaard, 1975). Layer thicknesses of 0.5 to 1 m observed in exposed vertical sides of cliffs near the ice margin suggested that a useful climatic record was captured in the ice cap (Mercer et al., 1975). Melting was observed at the margin, and from 5200 to 5380 masl penitente features suggested ablation (Hastenrath and Koci, 1981). Above 5400 m, surface albedo was approximately 80% during the austral winter, balancing net solar energy receipt and limiting ablation (Hastenrath, 1978). A 15-m core at the summit in 1976, ∼300 m above the equilibrium line altitude (ELA), captured the mass balance over 8 years from the annual layering of particles and oxygen isotopes (Thompson et al., 1979; Thompson, 1980).





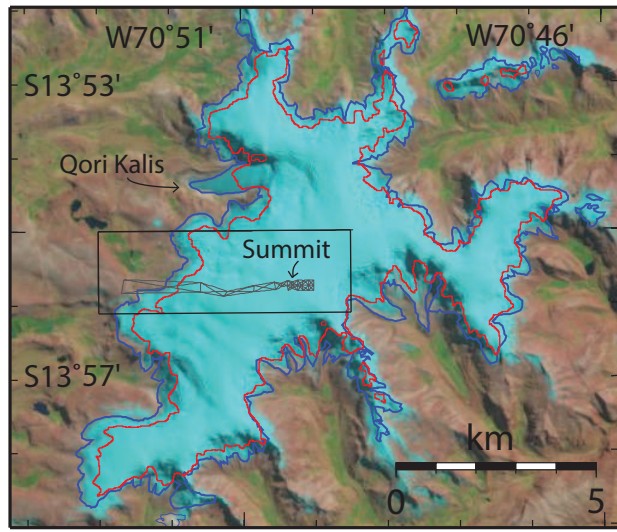

**Figure 1.** Landsat 5 TM image of the QIC on 20 June 1984 (LT50030701984172CUB01, available from http://eros.usgs.gov/). Ice extent in 1980 (blue line) and in 2010 (red line) derived from satellite images by Hanshaw and Bookhagen (2014). Survey network (black, braced quadrilaterals) in 1983 extended from bedrock to across the summit. Rectangle is region shown in Fig. 2 which includes a detailed view of the survey network.

Crevasse and pit sampling showed annual snow accumulation reductions of ∼30% were correlated with 5 El Nino records from 1964 to 1983, indicating the ice cap climatic record captured ENSO signals (Thompson et al., 1984). Ice thickness measurements suggested approximately 160-180 m ice thickness and flow modeling estimated an ice core of 600 to 1300 years length (Thompson et al., 1982). In 1983, existence of a millennium-scale climate record was confirmed with the recovery

of two cores in the summit region of the QIC, with one to bedrock at 163.6 m [Core 1] providing a 1500-year record of tropical precipitation in Peru (Thompson et al., 1985) and also capturing global climatic events including the Little Ice Age (Thompson et al., 1986). In 2003 a complete ice core was recovered and returned frozen to the laboratory (Thompson et al., 2013). Field studies continue on QIC and since 2003, a summit weather station has been maintained through annual servicing (Available at http://quelccaya.blogspot.com/), which has also afforded ongoing snow pit studies (Hurley et al., 2015) . In Oct.

2014, a 21-m-long ice core was collected at the summit.

Retreat of the Qori Kalis outlet glacier (see Fig. 1) between 1963 and 1978 was determined to be ∼70-100 m based on aerial and terrestrial photographs, respectively (Thompson et al., 1982). From an initial retreat rate of ∼5-6 m/yr from 1963 to 1978, subsequent terrestrial photographs in 1983, 1991, 1993, 1995, 1998, 2000 and 2005 showed increasing non-linear rate of retreat of the Qori Kalis margin — probably influenced by the formation of a pro- and sub-glacial lake — reaching ∼60 m/yr

by 2005 (Brecher and Thompson, 1993; Thompson et al., 2006). The photogrammetry also provided direct measurement of volume change and indicated ∼7-fold increase in the rate of volume decrease from between 1963-1978 to between 1983-1991.





**Figure 2.** (**A**) Detailed view of the 1983 survey network (gray, braced quadrilaterals). Red and blue lines as in Fig. 1. West ice margin in 2013 (yellow line) from GPS survey in the field. Horizontal surface velocity (blue arrows) from 1983 and 1984 field survey. Error ellipses (black) are 2-$\sigma$. Ice cores to bedrock in 1983 (black circle, Core-1, Thompson et al. (1985)) and 2003 (blue circle, Summit, Thompson et al. (2013)]). Survey monuments (black triangle, QSP-1) are on bedrock; established in the mid-1970s, they remain as of 2015. Elevations of the QIC surface were measured at the red triangles relative to QSP-1 in 1983/4 using terrestrial surveying and again in 2013 using dual-frequency GPS. Likewise, elevations at inverted red triangles were measured at six sites in 1978 and remeasured in 2015. (**B**) Measured change of the QIC surface elevation from 1983/4 to 2013 (red triangles) and 1978 to 2015 (inverted red triangles). Elevation decreased at all sites on the glacier over the 30- and 37-year spans. Five sites (circled) on the western margin are now exposed bedrock with over 60 m of ice lost at two sites. Thinning of the glacier surface extends across the summit where the average elevation decreased by 4.4 m. Error bars are 2$\sigma$.

Comparison of satellite imagery indicates the area of the entire QIC has decreased by 31% from 1980 to 2010, highly correlated with the change of area of Qori Kalis (Hanshaw and Bookhagen, 2014; Albert et al., 2014). In 2002, retreat of the west-central margin exposed rooted, soft-bodied plants approximately 5000 years old, suggesting recent conditions observed





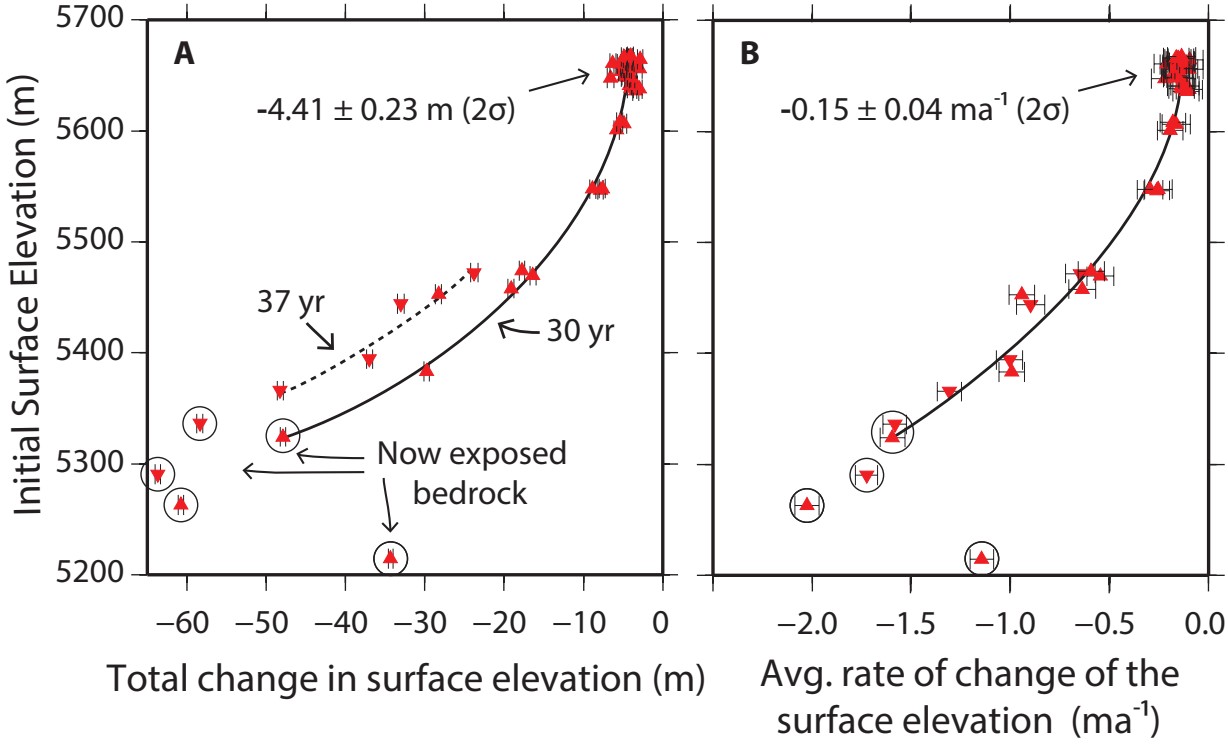

**Figure 3. A**) Total change in surface elevation from 1983/4 to 2013 (red triangles) and from 1978 to 2015 (inverted red triangles). **B**) Annual rate of elevation change, calculated by dividing values in **A** by span in years. The five lowest points (circled) are now exposed bedrock and thus both their total change and rate of change are minimum values. Error bars are $2\sigma$.

at QIC had not existed for about five millennia (Thompson et al., 2006; Buffen et al., 2009). Ice core records at the summit, which had captured the oxygen isotope record in 1983 deep cores, were no longer preserving the record by 1991, presumably due to meltwater infiltration at the summit (Thompson et al., 1993). This suggests a significant departure from conditions at the summit that had been in place over at least the previous ∼1500 years, as captured in long cores to bedrock.

5    To evaluate changes in the Quelccaya Ice Cap balance, we re-visited in 2013 and 2015 a survey network established in 1983 (Fig. 2A). As part of the field program that recovered the first deep ice cores from Quelccaya that year, a series of poles were placed in the glacier surface and surveyed to determine the horizontal strain rate at the summit, thus constraining a flow model for dating ice in the deep core. The network poles extended from the margin to the summit, encompassing both the ablation and accumulation zones. The elevation and horizontal displacement (from 1983 to 1984) were measured with decimeter precision

10   relative to two survey monuments on a rock outcrop west of the glacier (Fig. 2). While the poles on the glacier surface are gone, their locations (x,y) are known with better than sub-meter precision relative to the two monuments in rock. By re-occupying the same locations with geodetic GPS, any change in surface elevation since the initial survey could be measured. This provides





**Figure 4.** (**A**) Vertical profile of the QIC along the 1983 survey network from QSP-1 to the eastern margin. Surface elevations in 1983 (blue triangles and line) decreased at all sites by 2013 (red triangles and line). Bedrock location comes from gravity measurements (black diamonds, (Thompson et al., 1982)), mono-pulse radar (inverted brown triangles, Thompson et al. (1985)), ground penetrating radar (GPR, thick black line, Salzmann et al. (2013)), and length of cores to bedrock in 1983 (black, Core-1, Thompson et al. (1985)) and 2003 (blue, Summit, (Thompson et al., 2013)). Eastern portion of the profile (black triangles, dashed blue line, dashed brown line) is from a 1978 field survey (Thompson et al., 1982). The annual ice surface velocities (circa 1983-84, blue vectors) plotted in the vertical plane are shown without vertical exaggeration. (**B**) Ice unit-volume (area within profile x 1 m) was calculated from the flow divide at the summit to the western margin. The QIC unit-volume has decreased by 20 % over thirty years. Error bars, uncertainties, and ellipses are 2σ.

a geodetic mass balance for the transect up the ice cap (e.g., from 1983 to 2013); typically such geodetic mass balances are based on an old DEM from maps, stereophotogrammetric measurements or orthophotos, and a new DEM from LiDAR.





## 2 Data Collection

In late June 1983, 28 stations were laid out across the glacier from near the western margin to across the summit (Fig. 2). Stations consisted of ∼ 2 m long PVC poles secured 0.5 to 1.2 m into the snow, firn or ice, and protruded ∼1 m above the glacier surface. Placed with adjacent stations inter-visible, a single chain of braced quadrilaterals branched into two chains near the summit. In early July 1983, beginning at benchmarks 350 m apart drilled into bedrock west of the glacier, 77 distances were measured between adjacent poles using an Electronic Distance Measuring (EDM) instrument. The EDM used a near-infrared signal with a range of 2 km, and an accuracy of ±8 mm ($2\sigma$) as verified by calibration. To determine elevations, a one-arc-second least count theodolite was used to observe angles on both faces from both ends of lines, thus measuring so-called reciprocal vertical angles between adjacent poles. Reciprocal vertical angles were observed starting at bedrock benchmark QSP-1, then along the south side of the network through the doubly-wide chain of quads at the summit, returning along the north side and closing back on QSP-1. The mis-closure was -0.14 m.

Reciprocal vertical angles, corrected for deflection of the vertical (DOV), measure the ellipsoidal height difference between points. Beginning at QSP-1 the ellipsoidal height differences are added to calculate the ellipsoidal heights ($h$) at the pole tops. The orthometric heights ($H$) are calculated using

$$H = h + N, \tag{1}$$

where the geoid undulation, N, and the DOV are defined by the Earth Gravity Model 2008 (Pavlis et al., 2012). The length of the pole above the glacier surface was subtracted to estimate the surface elevation.

Four additional poles were placed near the west margin, south of the grid network and visible from QSP-1 (see the four most western, solid red triangles in Fig. 2). The positions at these four stations were measured directly from QSP-1 in July and again in early August 1983. In mid-August 1983, the 77 distances in the quadrilateral network were remeasured. Eleven poles across this network were then extended by 3 m and guyed in position to withstand winds and ∼3 m of snow accumulation expected during the 83/84 wet season.

In late August 1984, six of these extended poles were recovered intact. They were evenly distributed easterly along the network and a new temporary network of 14 additional poles was established around these six poles to allow remeasurement of distances and vertical angles to calculate updated positions. A total of 55 distances and reciprocal vertical angles were measured. Between the 1983 and 1984 poles, elevations were measured at 46 locations with a precision of ± 0.396 m ($2\sigma$) or better relative to QSP-1. EDM distances were reduced to UTM grid and adjusted (Wager et al., 1980; Chadwell, 1999) to estimate ice surface velocities plotted in the horizontal plane in Fig. 2A (and the vertical plane in 4A).

A 2012 reconnaissance found the bedrock benchmarks intact. Their global coordinates were measured in 2013 using geodetic-quality, dual-frequency GPS and calculated using NASA JPLs GIPSY software (Webb and Zumberge, 1997). Positions of the survey poles originally referenced to QSP-1 were transformed into global coordinates with 10 cm uncertainty. These coordinates were uploaded into a handheld GPS receiver and the dual-frequency GPS receiver was carried to within 1-2 m of each of the former pole locations. GPS data were collected at this spot and at four additional spots each about 5 meters





away along different orthogonal directions. Post-processing with GIPSY estimated the ellipsoidal heights with a precision of $\pm 0.30$ m ($2\sigma$). The ellipsoidal surface height at the precise location of the former pole was interpolated from a plane fit to the five points at and surrounding the site. The orthometric height was calculated using equation 1 with a precision of $\pm$ 0.36 m ($2\sigma$).

We checked for biases between the terrestrial and GPS survey by comparing measurements between the two bedrock control points. The EDM-measured distance was 349.982 $\pm0.008$ m ($2\sigma$) while the GPS-measured distance was 349.982 $\pm0.010$ m ($2\sigma$). The elevation difference was 3.446 $\pm0.036$ m ($2\sigma$) for the EDM/theodolite, and 3.448 $\pm0.030$ m ($2\sigma$) for the GPS measurements. Thus, there are no significant biases between the two measurement approaches.

Following the 2013 GPS survey and prior to the 2015 field season, a review of 1970s field logs uncovered six sites near the
western margin surveyed with terrestrial methods in 1978 from QSP-1 and -2. In 2015, the present-day elevations of the six sites were remeasured with GPS providing surface changes spanning 37 years.

## 3   Geodetic Results

The elevation decreased at all forty-six sites over the 30 years (Fig. 2B) and at six additional sites over 37 years. At thirty sites around the summit the average elevation decreased 4.41 $\pm$ 0.23 m ($2\sigma$). From the summit towards the western margin the
surface lowered with an increasing non-linear rate, with five sites at the western margin now exposed bedrock, with a maximum ice loss of 63.4 $\pm$ 0.34 m ($2\sigma$) at one site over 37 years (Fig. 3A). The average rate of thinning varies from -0.15 $\pm$ 0.04 ma$^{-1}$ ($2\sigma$) at the summit elevation of 5670 masl to -2.02 $\pm$ 0.06 ma$^{-1}$ ($2\sigma$) at 5260 masl in the lower ablation zone (Fig. 3B). For comparison, Brecher and Thompson (1993) observed the surface of Qori Kalis lowering 2.02 ma$^{-1}$ from 1983 to 1991 over an elevation range of ~4950 to 5150 masl.

Limited field time in 2015 allowed only two on-ice sites from 2013 to be re-observed. The 1983-2013 average annual surface change at the two sites was -0.79 $\pm$ 0.05 ma$^{-1}$ ($2\sigma$) while the 2013-2015 rate was -0.66 $\pm$ 0.15 ma$^{-1}$ ($2\sigma$), suggesting thinning continues unabated through 2015.

While the 1983-84 survey network only covers a portion of the area of the QIC, it does provide a vertical profile across the glacier from the margin to the summit (Fig. 4). Using a variety of ice thickness measurements made over the past thirty years,
the elevation of the bedrock beneath the profile can be established. By combining the surface elevations in 1983-84 with the bedrock elevations we estimate the unit-volume of ice along this profile to be 0.367 $\pm$ 0.003 x 10$^6$ m$^3$ ($2\sigma$). Comparing the ice surface elevation in 1983-84 to 2013, the unit-volume of mass lost was -0.078 $\pm$ 0.001 x 10$^6$ m$^3$ ($2\sigma$). This is a volumetric decrease from 1983 to 2013 of -21.2 $\pm$ 0.3% ($2\sigma$).

A geodetic mass balance estimate (Thibert et al., 2008) for the QIC is calculated by applying densities of 440 kg/m$^3$ above
5400 m elevation and 900 kg/m$^3$ below 5400 m elevation to the unit-volume mass lost equaling -0.057 $\pm$ 0.004 x 10$^6$ m$^3$ w.e. ($2\sigma$). Dividing by the unit-area of ~3900 m$^2$, the distance from the western margin to the flow divide times 1 m, gives the cumulative mass balance as -14.6 $\pm$ 2 m w.e. ($2\sigma$). This implies an average annual mass balance rate from 1983 to 2013 of -0.5 $\pm$ 0.1 m w.e a$^{-1}$ ($2\sigma$).





## 4  Continuity Relation

A glacier surface can lower in response to decreased mass balance, increased rate of densification/firnification, increased ice flux out of the accumulation zone, or decreased ice flux into the ablation zone. A change in surface elevation at any location with time, $\frac{\partial H}{\partial t}$, can be expressed through a vertically-integrated continuity relation as

$$\frac{\partial H}{\partial t} = \frac{\dot{b}_s}{\rho_s} - \frac{\partial q}{\partial x}, \tag{2}$$

where $\dot{b}_s$ is surface specific mass balance rate, $\rho_s$ is the density of the surface material, and $\frac{\partial q}{\partial x}$ is the ice flux divergence along the flow line (Whillans, 1977). Here we have assumed two dimensional flow, i.e., no transverse flow and negligible basal mass balance.

The annual surface specific mass balance rate ($\dot{b}_s$) can be calculated as

$$\dot{b}_s = \dot{a}_s - \dot{m}_s + \dot{a}_r - \dot{s}, \tag{3}$$

where $\dot{a}_s$ is snowfall, $\dot{m}_s$ is melt, $\dot{a}_r$ is refreezing, and $\dot{s}$ is sublimation, all as annual rates. Melt and sublimation are dominant processes within the ablation zone and a sustained increase in their magnitudes will decrease $\dot{b}_s$, lowering the surface elevation. In the accumulation zone, any melt likely refreezes; therefore, the net value of $\dot{b}_s$ remains unchanged, though density ($\rho_s$) increases, effectively lowering the surface elevation. A decrease in snowfall or increase in sublimation directly lowers the surface elevation.

Ice flux divergence is the difference between the ice flux entering and exiting laterally through opposite faces of a vertical column. An increase raises the ice surface, while a decrease lowers it. Lacking direct measurements of the horizontal motion of ice all along the vertical faces of the column, a good proxy is measuring the vertical component of ice motion at the surface, the emergence velocity. Assuming negligible basal motion, the ice flux divergence is given as

$$\frac{\partial q}{\partial x} = -(w_s - u_s \frac{\partial S}{\partial x}), \tag{4}$$

where $w_s$ is the vertical component of the ice velocity, $u_s$ is the horizontal component of ice velocity and $\frac{\partial S}{\partial x}$ is surface gradient $= tan(\alpha)$, where $\alpha$ is the surface slope. Substituting Equation 4 into Equation 2 gives the change in surface elevation over time through the continuity relation (Paterson, 1994) in terms of the emergence velocity as

$$\frac{\partial H}{\partial t} = \frac{\dot{b}_s}{\rho_s} + (w_s - u_s \frac{\partial S}{\partial x}). \tag{5}$$

The emergence velocity in 1983-84 (Fig. 5) is calculated from the components of ice surface velocities and the local slope of the glacier surface at each site shown in Fig. 4A. For the lower three points the uncertainties are larger because the survey spanned one month in 1983 and values are extrapolated to an annual rate, and due to uncertainties in measuring the local slope of the ice surface around the stake/pole sites in 1983. At sites above 5400 masl, stakes/poles were recovered intact in 1984 and the emergence velocities are calculated over a full year.



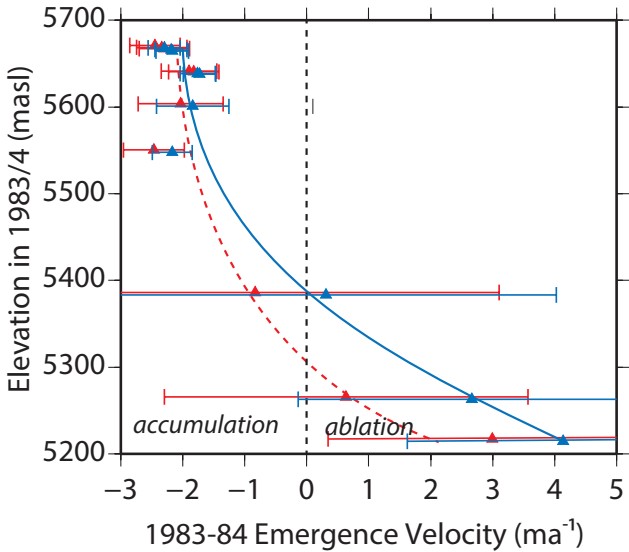

**Figure 5.** Emergence velocity calculated from observations in 1983 and 1984 and plotted versus elevation (blue, solid line). The larger error bars below ∼5400 m are due to extrapolating observations spanning one month in 1983 to an annual rate. Emergence velocities are a measure of ice flux divergence. Negative velocities move mass downward and out of the accumulation zone while positive velocities move mass into and upward in the ablation zone. The velocities are balanced (zero) at the ELA that was ∼5400 m in 1983-1984. Emergence velocity (red, dashed line) that would account for the 1983 to 2013 thinning suggests an ELA lowering of ∼75 m, assuming no change in mass balance rate nor density. Elevations offset by 5 m for clarity. Error bars are ($2\sigma$).

## 4.1 Application to the 1983-84 Data

Between 1983 and 1984, the specific mass balance was measured at nine stakes, and along with their emergence velocities (Fig. 5) are used in the continuity relation (Equation 5) to calculate any change in surface elevation. At the summit, measurement of the specific mass balance over a 414-day period, following an El Nino, from July 1983 to August 1984 was 0.92 ±0.14 m w.e. ($2\sigma$), which closely agrees with the average specific mass balance rate of 0.87 m w.e. during El Nino years (Thompson et al., 1984). At the summit, the displacement by the emergence velocity over the 414 days was -2.60 ± 0.26 m ($2\sigma$). Using an average density of 430 kg/m$^3$ from mid-1970s snow pits (Mercer et al., 1975; Thompson, 1980) the continuity relation calculates a lowering of the summit elevation of -0.51 ±0.28 m ($2\sigma$) while direct observations of the surface elevations relative to the bedrock monument QSP-1 show a lowering of -0.48 m ±0.32 m ($2\sigma$). This demonstrates the applicability of the continuity equation to relate surface elevation, mass balance rate and emergence velocities. Furthermore, the cause of the lowering from 1983 to 1984 can be attributed directly to reduced mass accumulation, presumably due to enhanced ablation following the 1983 field season.

The continuity relation is next used to estimate a specific mass balance rate profile under steady-state conditions in 1983-84. Steady state implies that no surface lowering occurs, which is equivalent to setting $\frac{\partial H}{\partial t} = 0$ in Equation 5. Using the measured





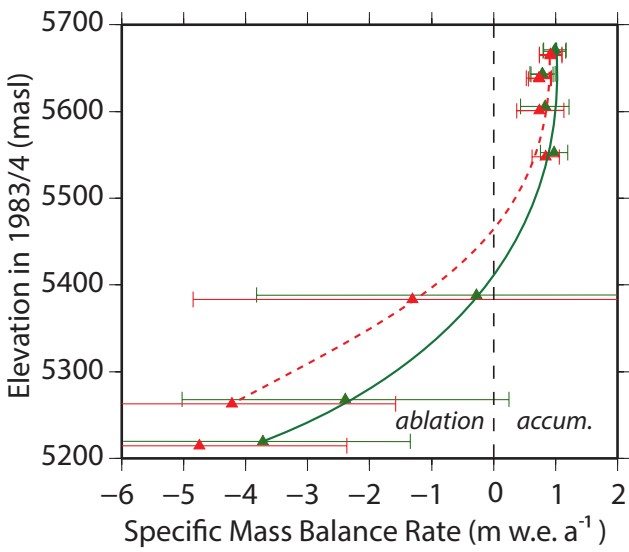

**Figure 6.** Geodetically-derived specific mass balance rate profiles for 1983-84 steady state conditions (green, solid line) and for 1983-2013 average rate that accounts for the observed thinning (red, dashed line) assuming no change in emergence velocities or densities; ELA increase is ~50 m. Elevations offset by 5 m for clarity. Error bars are ($2\sigma$).

emergence velocities, Equation 5 is solved for $\dot{b}_s$ using densities of 440 and 900 kg/m$^3$ in the accumulation and ablation zones, respectively (Fig. 6).

## 4.2 Application to the 1983-2013 Data

Changes in mass balance, firn density and ice flux are theoretically all possible explanations for the thinning of QIC between
5 1983 and 2013. To investigate, the thinning values observed from 1983 to 2013 are introduced into the continuity relation by setting $\frac{\partial H}{\partial t}$ equal to the thinning rates shown in Fig. 3B. Assuming the steady-state mass balance rate and emergence velocities from 1983-84, and densities described above, two of these three are held fixed and the third solved for to account for the entire thinning rate. The required emergence velocity is shown in Fig. 5, the required mass balance rate in Fig. 6, and the required densities within the accumulation zone in Fig. 7. These results are discussed in the following.

## 10 5 Discussion

The observed change in surface elevations on QIC is similar to those seen at other glaciers. The non-linear increasing rate of thinning with lower elevations has previously been observed at mountain valley glaciers in the northern hemisphere (Schwitter and Raymond, 1993). While ice margin retreat has been observed at tropical Andean glaciers (Vuille et al., 2008) and tropical glaciers in general (Kaser, 1999), there are few repeated measurements of surface elevations and volumes on tropical glaciers. Where studies
15 have been conducted thinning and volume loss are the dominant trends.





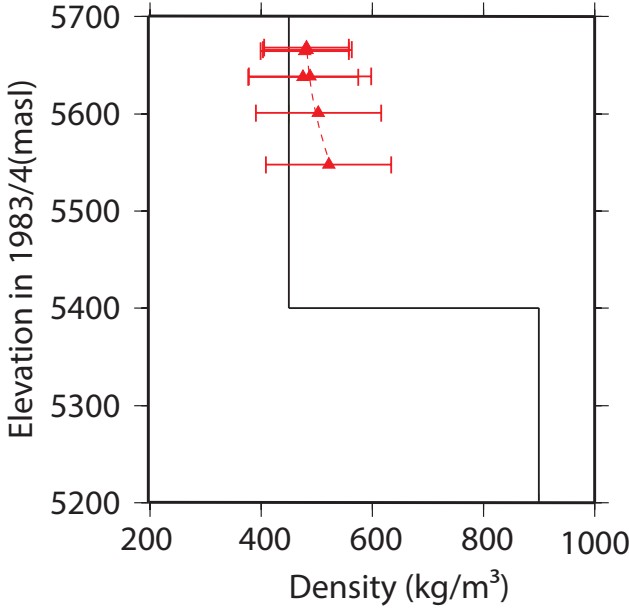

**Figure 7.** Initial (black, solid line) densities are 440 and 900 kg/m$^3$ in the accumulation and ablation zones, respectively. Calculated (red, dashed line) increase in densities within the accumulation zone to account for observed thinning assuming no change in steady-state mass balance and emergence velocities from 1983-84. Within the ablation zone, ice is already at 900 kg/m$^3$ and no practical increase in density can account for the surface lowering. Error bars are ($2\sigma$).

In Cordillera Blanca of Peru, Hastenrath and Ames (1995) observed from 1977 to 1988 that the Yanamarey Glacier, with a maximum elevation of ~5100 m, thinned about 3 m/yr in the ablation zone, with a geodetic mass balance rate of -1.5 m w.e. a$^{-1}$. Mark and Seltzer (2005) estimate a geodetic mass balance rate of -0.35 m w.e. a$^{-1}$ from 1962 to 1999 for glaciers on the Nevado Queshque which has a maximum elevation of ~5600 m. In the Cordillera Raura of Peru, Ames and Hastenrath

(1996) find a geodetic mass balance rate of -2 m w.e. a$^{-1}$ from 1977 to 1983 on the Santa Rosa Glacier which has a maximum elevation of ~5400 m. Soruco et al. (2009) observed volume decrease of 43% at 21 glaciers in the Cordillera Real in Bolivia from 1963 to 2006 with geodetic mass balance rates ranging from -0.26 to -1.38 m w.e. a$^{-1}$. Rabatel et al. (2013) suggest that generally glaciers in the Andean tropics have mass balance rates of -0.6 m w.e. a$^{-1}$ if the maximum elevation exceeds 5400 m and -1.2 m w.e. a$^{-1}$ if the maximum elevation is less than 5400 m. Mass loss at Quelccaya with a maximum elevation of

5670 m and with a geodetic mass balance rate of -0.5 ± 0.1 m w.e a$^{-1}$ ($2\sigma$) is consistent with that measured at other Andean glaciers.

Outside the tropical Andes, observations of ice surface elevation on Kilimanjaro over two different short intervals (31 and 16 months) in the mid-2000s find a lowering between 0.25 to 0.65 ma$^{-1}$, respectively (Pepin et al., 2014). Over a 38-year period from 1962 to 2000 photogrammetric comparison yields an average thinning of ~0.5 m a$^{-1}$ of Kilimanjaro's ice surface

Thompson et al. (2002).

We next examine in detail the processes occurring within the ablation and accumulation zones on Quelccaya.





## 5.1 Ablation zone

Comparison of mass balance rates under steady state and thinning conditions in Fig. 6 suggests that below $\sim$5400 masl in the ablation zone, mass losses of 1 to 2 m w.e. a$^{-1}$ can account for the observed thinning and margin retreat. A likely contributing factor is increasing air temperature at QIC, on the order of 0.1 to 0.3 °C/decade over the past 30-40 years based on Reanalysis trends (e.g., Bradley et al. (2009); Schauwecker et al. (2014)) and extrapolating station data trends (Fig. 5 of Vuille et al. (2015)). Increased ablation is also supported by an approximate relationship between rate of margin retreat $\frac{\partial L_a}{\partial t}$ and ablation as

$$\frac{\partial L_a}{\partial t} = \dot{a}_i \frac{L_a}{H_a}, \tag{6}$$

where $\dot{a}_i$ is the perturbation in ablation rate from steady state, $L_a$ is nominal length of the ablation zone, $H_a$ the nominal ice thickness of the ablation zone (Cuffey and Paterson, 2010). The west margin of QIC has retreated approximately 300 m in 30 years giving $\frac{\partial L_a}{\partial t} = 10$ ma$^{-1}$. With $L_a = 1000$ m and $H_a = 100$ m (See Fig. 4) this gives $\dot{a}_i = 1$ ma$^{-1}$, consistent with that shown in Fig. 6.

Alternatively, the observed thinning (and retreat) of QIC could be due to a decrease in ice flux into the ablation zone (See Fig. 5). The nominal response time ($t_r$) for a change in mass balance rate in the accumulation zone to affect the margin is approximately

$$t_r = \frac{H_o}{\dot{a}_o}, \tag{7}$$

where $H_o$ is the nominal thickness of the glacier in the ablation zone and $\dot{a}_o$ is the nominal ablation rate at the margin (Johannesson et al., 1989). Taking $H_o = 100$ m and $\dot{a}_o = 4$ ma$^{-1}$ gives $t_r = \sim$25 years.

As recorded in the 2003 core at the summit, the average annual mass balance rate was 1.18 m w.e. a$^{-1}$ from 1883 to 1983 (Thompson et al., 2013). Given this is $\sim$4 times longer than the nominal glacier response time, this suggests that the ice flux into the ablation zone had likely been steady during the 100 years preceding the onset of rapid margin retreat in the mid-1980s. The significant mass losses we have documented since the mid-1980s would generally reduce the driving forces and slow ice flux into the ablation zone (Kruss and Hastenrath, 1983; Rabus and Echelmeyer, 1998). This would accentuate thinning within the ablation zone and margin retreat, and is generally consistent with reduced emergence velocities as shown in Fig. 5 required to account for the thinning. Alternatively, increased basal sliding through melt-water lubrication would increase ice flux, though thinning and margin retreat could continue, if melt and sublimation rates accommodate the increased ice flux. These possibilities can only be resolved by remeasuring the surface velocities at the same locations as in 1983 to detect changes in ice flux.

Slowing ice velocities influencing surface elevations and mass balance in general, has been observed. Hastenrath (1987) found a decrease of about half in the surface velocities of the Lewis Glacier, Mount Kenya between 1978 and 1985 due to decreased mass balance. Berthier and Vincent (2012) found that more than two-thirds of increased thinning rate in the lower



Mer de Glace, French Alps, is caused by a ∼50% decrease in ice velocity from 1978 to 2008. At several sites distributed globally, glacier slowing has been observed by repeat optical satellite imagery (Heid and Kaab, 2012).

## 5.2 Accumulation zone

As noted, thinning in the accumulation zone can result from lower surface mass balance, increased firnification rate, increased ice flux from the summit, or a combination thereof. The geodetically-derived, steady-state mass balance rate at the summit for 1983-1984 was 1.04 ±0.18 m w.e. a$^{-1}$ ($2\sigma$). We postulate that any change in mass balance rate at the summit occurs gradually, and is not a step change. Assuming ice flux and firn density are unchanged at the summit, a linear change of only -0.0044 m w.e. a$^{-1}$ per year acting from 1983 to 2013 could account for the observed lowering of 4.41 ± 0.23 m ($2\sigma$). This would reduce the geodetically-derived annual mass balance rate to 0.91 m w.e. a$^{-1}$ by 2013.

The surface elevation can also lower due to melting and refreezing accelerating firnification (Bezeau et al., 2013). As shown in Fig. 7, the average density of each year's new accumulation need only increase from the initial value of 440 kg/m$^3$ to a trend of 470 to 510 kg/m$^3$ with lower elevation or higher freezing level to account for the total surface lowering. We next examine the snow pit and ice core data for changes in accumulation and density.

Snow pit sampling has been a key component of Quelccaya research since 1974, allowing evaluation of accumulation through time. Measurements of depth and density in June or July allow annual accumulation to be expressed as water equivalent, representing the integration of wet season snow accumulation — typically with an onset in early October — with ablation processes both during the wet season and until the time of sampling. Although they do not reflect processes occurring through the balance of the dry season (∼June/July to early October), and therefore cannot represent true annual mass balance, annual snow pit accumulation values can be directly compared.

Summit snow pit data recorded an average accumulation rate of 1.10 ± 0.12 m w.e. a$^{-1}$ ($2\sigma_{mean}$, n=8) from 1976 to 1983 (Thompson et al., 1984). Summit snow pit data from 2007 to 2015 recorded an average accumulation of 1.02 ±0.10 m w.e. a$^{-1}$ ($2\sigma_{mean}$, n=9), results updated from Hurley et al. (2015). This is a linear change of $-0.0027$ ±0.0026 m w.e. a$^{-1}$ per year ($2\sigma$) over 30 years and amounts to -2.7 ± 5.3 m ($2\sigma$) of elevation change at the summit. From 1983 to 2002, the 2003 ice core recorded an average mass balance 1.16 ± 0.20 m w.e. a$^{-1}$ ($2\sigma_{mean}$, n=20) with a linear trend of $-0.0054$ ±0.0083 m w.e. a$^{-1}$ per year ($2\sigma$) (Thompson et al., 2013). Extrapolating this linear rate to 30 years amounts to -5.5 ± 15.8 m ($2\sigma$) of elevation change at the summit.

Snow pits at the summit from 1974 and 1976-1978 had an average density of 430 ±46 kg/m$^3$ ($2\sigma_{mean}$, n=4) (Mercer et al., 1975; Thompson, 1980), while the average snow pit density at the summit from six years between 2008 and 2015 was 441 ±22 kg/m$^3$ ($2\sigma_{mean}$, n=6), updated from Hurley et al. (2015). While this is not a statistically significant change, comparison of density profiles of the upper 0-25 m of the 1983 and 2003 ice cores (Thompson et al., 1985, 2013) also suggests an increase in the firnification rate. Assuming a linear increase in density from 430 to 441 kg/m$^3$ would amount to summit elevation change of -0.9 ± 4.4 m ($2\sigma$) over 30 years.

The geodetically-derived annual mass balance rate of 1.04 ±0.18 m w.e. a$^{-1}$ ($2\sigma$) — representing only one year — is not significantly different than values derived from snow pits and the 2003 ice core. Both pit and core data show decreasing linear





trends that amount to $2.7 \pm 5.3$ m ($2\sigma$) and $5.5 \pm 15.8$ m ($2\sigma$) of summit elevation lowering, respectively. Adding the $0.9 \pm 4.4$ m ($2\sigma$) lowering due to density increase, gives a total summit elevation lowering of $3.6 \pm 7$ m ($2\sigma$) and $6.4$ m $\pm 16$ m ($2\sigma$), from the pit and core records, respectively. While these magnitudes are comparable to the observed summit lowering of $4.41 \pm 0.23$ m ($2\sigma$), the glaciological measurements' resolution and high degree of interannual variability do not yet allow a

statistically meaningful determination of the extent to which decreased mass balance and/or increased density contribute to the observed lowering.

Finally, increased ice flux out of the accumulation zone could also lower the ice surface elevation (See Fig. 5). For example, under steady-state conditions for mass balance rate and density, the emergence velocity at the summit would have to decrease from $-2.30$ to $-2.45$ m a$^{-1}$ to account for the observed thinning. As noted, the significant mass loss in the ablation zone

would generally reduce the driving forces and slow ice flow and because this has been underway since the mid-1980s - a span comparable to the glacier response time - the QIC may have adjusted (slowed) the ice flow rates to balance the masses. However, predicting changes in ice flow rate is complicated. It is an interplay between flow being proportional to the fifth power of ice thickness, which has decreased, and flow proportional to the third power of slope, which has increased (Nye, 1952). Additionally, an alternative mechanism could be increased basal sliding through melt-water lubrication (e.g., Sole et al.

(2008)). Again, these possibilities can only be resolved by remeasuring the surface velocities at the same locations as in 1983 to detect changes in ice flux.

## 6  Conclusions

The summit elevation of QIC has lowered by $4.41 \pm 0.23$ m ($2\sigma$) over thirty years or an average of $-0.15 \pm 0.04$ ma$^{-1}$ ($2\sigma$). The surface elevation decreased at an increasing rate towards the western margin, with a maximum ice loss of $63.4 \pm 0.34$

20  m ($2\sigma$), accompanying retreat of the western margin by $\sim$300 m. The geodetic data span 30 years, which approximates the nominal response time of Quelccaya, and likely captures its longterm response rather than just interannual variability in snow accumulation and melting. The unit-volume of ice lost along a profile from the margin to the summit from 1983 to 2013 was $20.4 \pm 0.3\%$ ($2\sigma$). The cumulative mass balance along this profile was $-14.6 \pm 2$ m w.e. ($2\sigma$) implying an average annual mass balance rate of $-0.5 \pm 0.1$ m w.e. ($2\sigma$) for the QIC.

Within the ablation zone, surface lowering is caused by a combination of melt, sublimation and ice flux rate leading to an effective ablation rate of 1-2 m w.e. a$^{-1}$. Within the accumulation zone, annual snow accumulation recorded in summit pits likely decreased, and bulk density likely increased, contributing to the observed surface lowering, however, these observations are not yet statistically significant. Increasing air temperature is probably responsible, accelerating melt and sublimation in the ablation zone, and firnification and percolation of meltwater in the accumulation zone during the period of this study. Precise

resolution of cause(s) will require continued annual glaciological measurements. The role of changes in ice flux remains unconstrained until ice flux divergence (emergence velocity) can be remeasured across the Quelccaya Ice Cap.



*Acknowledgements.* Resurvey funded by UC Academic Senate Grant RK-030S and National Geographic Grant 9132-12 ; AWS and snow measurements supported by NSF-P2C2 (AGS-1303828), NSF Paleoclimate (9909201 and 0402557), and the NOAA Global Climate Observing System. Phillip Kruss guided the network layout in 1983 and assisted with the data collection. Keith Mountain assisted with the 1984 survey. Toby Meierbachtol collected the Ground Penetrating Radar data and reprocessed it for this paper. We thank guides and staff of

5  Vicencio Expeditions for logistical support in the field. Figures prepared using Generic Mapping Tools (Wessel and Smith, 1998).





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
