# Peer review of "Thinning of the Quelccaya Ice Cap over the last thirty years"

_The Cryosphere, 2016_

## Referee Comment (RC1) · Anonymous Referee #1 · 15 Apr 2016

Title: Thinning of the Quelccaya Ice Cap over the last thirty years Manuscript: The Cryosphere Discuss., doi:10.5194/tc-2016-40, 2016 Authors: C.D. Chadwell, D.R. Hardy, C. Braun, H.H. Brecher, and L.G. Thompson

General Comment:

This manuscript provides detailed information about surveying work conducted on the Quelccaya Ice Cap (QIC), which allows the authors to report on thinning and volume loss along a profile of the western part of the ice body over the last 30 years. The recent field work took place in 2013 and 2015, and was compared to a network established in 1983 – the same year the first deep ice cores were recovered from the QIC. The main contribution of this research is to present up to date numbers about thinning and the volume changes that have taken place since the early field work was completed,

with the core of the manuscript describing the data collection and surveying methods. A continuity relation is also presented, with the purpose of providing information about the physical processes controlling the observed changes. These findings are a little more tenuous than the surveying results, which is evident in the manner in which they are described by the authors. A decision should probably be made as to whether these latter results are really necessary and/or if they really add to the understanding of the physical processes controlling the observed retreat. However, if they are not included there is a question mark over whether the surveying results on their own are enough in their present form to warrant publication. To aid in that decision the following comments about the manuscript are offered, which the authors may wish to consider should the paper be considered for publication in The Cryosphere.

Specific comments:

Please note that page number is referred to as (P) and line number is referred to as (L).

1. Structure: The overall structure of the manuscript could be refined to fall within a more conventional format. There is no clear delineation between methods-results-discussion in the manuscript in its present form. The aim and/or objectives should be more clearly defined at the end of the introduction and it would be valuable to provide information about how the manuscript is structured. The second section describes the data collection, which is followed by geodetic results (section 3). One could argue that the continuity relation and the methods used to characterise the physical processes in the ablation and accumulation zones (Sections 5.1 and 5.2 in the discussion) could be contained in a discrete methods section. The present discussion (Section 5) is not really a synthesis of all the results. The manuscript would be strengthened if a clearer methods section was defined, followed by key results and a synthesis of the importance of these in relation to previous and ongoing research.

2. Abstract: The first part of the abstract is generally well written and provides information about the thinning and volume loss of the "unit-volume" (that needs further clarification in the manuscript – see below). May one ask why the first sentence refers specifically to the challenges of acquiring data from "accumulation zones" of tropical glaciers when the results represent changes in elevation along a transect of QIC? There is no clear definition of accumulation or ablation zones in the manuscript, and no information is given about the areal extent of each of these zones on QIC. The latter part of the abstract, starting with the sentence "increasing air temperatures" and those that follow are not strongly supported by the results contained in the manuscript. In particular, the statement about increasing air temperature being "a major driver of the observed changes" is not demonstrated by results presented in the manuscript. The sentences that follow are also quite "speculative" and not really supported adequately to be deemed as results. There is no insight as to what "steady-state" refers to and the causes of the surface lowering in the accumulation zone are questionable.

3. Introduction: The introduction is very specific to QIC and does not provide any context for the broader motivation for the research. The first paragraph is very detailed and would almost be better suited as a site description in the methods. If retained, then the area of QIC should be provided here, or elsewhere, as reference to QIC being the "largest tropical glacier" is not really adequate. The failure to provide the area in the manuscript prevents readers from fully appreciating what the "ice unit-volume" represents. As noted above, a clearer aim and some structural information about the manuscript would be of benefit to readers at the end of the introduction.

4. Data collection: This is detailed and provides readers with the necessary background about the approach. It might be of interest to reveal to readers what parties were responsible for the early measurements versus those responsible in 2013 and 2015. On P6, L7 the accuracy of the EDM instrument is described, which is the first of many numbers in the manuscript that contain an uncertainty. It would be useful to provide additional information about how this and other uncertainties are calculated. All the uncertainties in the manuscript are given as 2 standard deviations ($2\sigma$) from

the value presented, but it is difficult to know in many instances how these are actually calculated. Some effort to unpack these, especially for the surveying results would be useful to readers to help strengthen the findings presented. A table that provides a summary of the field measurements, including the number of stakes installed versus those re-measured and/or surveyed, with the dates when this occurred would help support the text provided to describe these important activities.

5. Geodetic results: As noted above, additional information about how uncertainty is calculated in the elevation change results would be useful to readers. To ensure that the volume changes are more meaningful, the authors should consider more clearly defining what portion of the QIC is represented by the 1983-84 survey network (P7, L23-24). "Using a variety of ice thickness measurements made over the past thirty years" is not very clear and the "unit-volume of ice along this profile" should be constrained in some way, otherwise the volumetric decreases are not that meaningful. The uncertainty in the annual mass balance is shown to be $\pm$ 0.1 m – again, how is this calculated and does it take into account, for example, uncertainty associated with applying specified snow densities?

6. Section 4: The use of the continuity equation to relate surface elevation, mass balance and emergence velocities (P9, L6-12) is justified by the close comparison of the measurements of surface lowering of the summit elevation and those calculated. How confident are the authors that the similarity of the presented means justify the use of the continuity equation to unravel the key physical processes in the ablation and accumulation zones? The uncertainty in these estimates is in the same order of magnitude of the values determined. Are they really that similar and how sensitive are they to the assumptions necessary to derive them?

7. Discussion: The description of the physical processes occurring in the ablation and accumulation zones (sections 5.1 and 5.2) do not read like a "discussion" section. Both sections are quite data rich and some of the data presented fall outside of the previously defined study period (e.g. Section 5.2 (P13, L15-33 – reference to snow pit

sampling starting in 1974). The results presented in Section 5.1 do support the notion that increased ablation can account for the observed thinning and margin retreat, but this section does not provide enough evidence to show what physical processes are responsible. In the abstract, it is stated that it is "melting and sublimation above steady-state" – differentiating between these two is extremely important if air temperature is deemed responsible, which is not done in Section 5.1. The physical processes responsible for surface lowering in the accumulation zone remain poorly constrained using the approach adopted (Section 5.2) – the lack of atmospheric observations or application of a climate model (e.g. energy and mass balance modelling) prevents the authors from adequately distinguishing between the importance of changes in accumulation and densification of snow and ice (firnification rate) – acknowledged by the authors (P14, L3-5).

8. Conclusions: The first paragraph and the results presented in relation to surface lowering and volume loss are of interest. However, the second paragraph (like the abstract), starting with the statement that surface lowering is caused by a combination of "melt, sublimation and ice flux rate" does not provide any new insight – one would expect surface lowering to be a combination of these over most retreating glaciers. It raises the question as to how useful the continuity relation is to this research and if the latter part of the research is that valuable. The approach reads a little bit like a "back of the envelope" exercise – it succeeds in providing a framework to assess the possible causes of thinning and volume loss but ends up failing to fully account for the physical processes responsible. The same applies to the results from the accumulation zone – again, the authors acknowledge that the observations are not "statistically significant". If the latter part of this research is to be retained, it requires further justification as to why it is useful and how it provides a building block to ongoing research. Some explanation as to why climatological data are not included to help resolve some of the uncertainty is almost necessary.

9. Final comment: The observational data is of interest to readers given the challenges

to extract such valuable information from tropical glaciers – especially the largest one remaining on earth. However, the manuscript falls a bit short in adequately accounting for the physical processes responsible for the observed changes. It succeeds in showing some of the likely mechanisms responsible for the observed changes but is this really enough?

Minor technical suggestions

P1, L1: small t for tropical glaciers P2, Figure 1: It would be useful to have an insert that shows where QIC is in relation to the regions that surround it.

---

## Referee Comment (RC2) · Anonymous Referee #2 · 18 Apr 2016

The authors provide data on the thinning rate and volume loss (per unit area of transect) of the world's largest tropical ice cap, the Quelccaya Ice Cap (QIC). They provide support from the literature for the values observed thinning rates and modeled mass balance changes. These data are a significant contribution to our current knowledge about tropical glacier evolution over the past few decades, in a region where such measurements are limited. Additionally, the authors use the continuity relationship to evaluate various [climatological and dynamical] causes for the observed thinning at the QIC. Their conclusions about causes for observed thinning (mainly increased ablation rates in the ablation zone and a possible combination of density changes and changes in accumulation in the accumulation zone) are supported by their methods, data, and applications of previous research. The authors also outline areas where additional research could strength the results of their study and the field's understanding of the QIC

responses to climate change, namely whether ice velocities have changed since the measurements in the mid-1980s. The paper is well thought out and beneficial to the field, but I have some broad and specific comments on the content and presentation:

I suggest re-formatting on the Introduction. The Introduction provides a detailed and chronological list of past work at the ice cap. The driving problem and specific questions of the research, however, are not provided in the Introduction.

For the application of the continuity equation to enlighten the causes of the QIC thinning, there were instances where I was not clear on the assumptions being made. Below, in the Specific Comments section, I enumerate these occurrences.

Additionally, much of the results hinge on whether mass fluxes from the ablation zone to the accumulation zone have changed, which the authors outline as an area for further research. Are there end-member cases (on ice velocity) that that the authors could explore to quantify how dependent their conclusions are on whether mass fluxes have changed? By adding these end-member scenarios, their conclusions on the causes of observed thinning would be stronger.

The authors provide a valuable data set on ice thinning and mass loss at the QIC and that will aid future research of low-latitude glaciers. The writing and figures are clear and logical, and the paper is well supported by the literature. If the authors address the broad comments (above) and specific comments (below), then I would recommend this manuscript for publication.

Specific Comments:

pg. 1, line 2: should you cite a reference on the QIC being the largest tropical ice mass?

Figure 1/Figure 2: Can these figures be combined? I found the quadrilateral network in Figure 1 seems to be important in Figure 2 but slightly distracting in Figure 1 as a stand-alone figure.

pg., 2, line 16: Is there a citation for the increasing rate of volume loss?

pg. 5, lines 1 & 2: I'm not sure what this sentence about how 'typically such geodetic mass balances are based' is trying to say. Are your methods A-typical? By ending the introduction this way, I'm left wondering why this hasn't been done before, rather then excited that this paper will fill in the gap. This specific comment ties back to the first general comment.

pg. 5, line 7: Has the EDM data from 1983 not been published before?

pg. 7, lines 29 & 30: What are the implications of using a step-wise density profile? From Figure 7, I see that the density profile does affect the modeled results on thinning rate. Has a sensitivity analysis been conducted on the assumptions about the density profile?

pg. 8, lines 1 & 2: The last two listed possible causes for glacier surface lowering seem to contradict one another. If there is an increased flux out of the accumulation zone, must there also be an increased (instead of decreased) flux into the ablation zone; correct? Also, with tropical glaciers, which have large accumulation area ratios, a decreased flux into the ablation zone should mean that more mass stays in the accumulation zone, which encompasses a greater area, and thus increasing the total height of the glacier. Am I missing something?

pg. 8, line 19: Is it appropriate to assume negligible basal motion? The ice core records indicate that the QIC is not a cold-based glacier. Thus basal sliding is permissible. I am not sure if previous work has attempted to constrain the sliding versus deformational velocity of the ice cap. How substantial does basal sliding need to be before this assumption is invalid?

pg. 9, line 14: What information does assuming a steady-state height in 1983-1984 provide? This seems to be a key point of your analysis of the thinning rate measurements. Also, how reasonable is this assumption? The Qori Kalis outlet glacier has retreated

throughout the direct observational and photographic record. And from Figure 2, in the region of the survey, the ice retreated even between 1980 and 1984. These observed retreats would lead me to believe that the ice cap was not in equilibrium in 1983 – 1984 with the climate. Of course, assumptions must be made when using models, but what are the reasons for these assumptions and what are possible implications?

Figure 5: Is it a coincidence of the data that the emergence velocity is zero at ∼5400 m in 1983-1984 (blue curve)? Or did you prescribe the curve with that value, since that is roughly the ELA from snow line measurements in the late 1970s? Also, what does a lowering of 75 m in the ELA accounting for the observed thinning mean? I don't see this point discussed in the text.

Figure 6: Is it appropriate to think of the red dashed line as the specific balance profile today? And is the difference between the two curves is the mass balance anomaly per elevation? How does the 50 m increase in the ELA compare with values in the literature?

pg. 10, lines 13 & 14: This point should also be made in the introduction and the abstract.

pg. 12, line 11: How do you determine the perturbation in the ablation rate? Are you assuming that in Figure 6, the perturbation in the ablation rate is the difference between the green solid and red dashed curve? At ∼5400 m a.s.l., that difference looks to be ∼1 m w.e. per a, but at ∼5300 m a.s.l. (near the ice margin) that difference looks to be closer to 2 m w.e. per a, which is order unity and reasonable for Equation 6. But in the text or the figure captions a bit more information about what the curves mean would be helpful.

pg. 13, lines 6 - 9: These assumptions seem inconsistent with the ice core record and influence of ENSO. If I plot the 2003 South Dome ice core record (plus recent snow pits) (https://www.ncdc.noaa.gov/cdo/f?p=519:1:::::P1_study_id:14174) from thermal year 1976 through thermal year 2009 and plot a linear regression, I find a net summit ac-

cumulation rate of change of +0.001 m w.e. per year ) (r-squared value of 0.001 and p-value of 0.81). If I regress from thermal year 1984 through thermal year 2009, the net summit accumulation rate of change is + 0.002 m w.e. per a (r-squared value is 0.001 and p-value is 0.84). If I were to just draw a line from the thermal year 1983 to the thermal year 2009, then I see a net summit accumulation rate decrease of -0.018 m w.e. per a, but this method would not capture a trends in the accumulation. During this period, the 1-sigma value of the net annual summit accumulation is ∼0.4 m w.e. Even if the starting and ending summit mass accumulation values (that I draw the line between) were several-year averages, connecting dots would only be valid if those averages had similar statistics on ENSO events.

pg. 14, line 26: Is 'effective ablation rate' a perturbation in the ablation rate compared to 1983-1984?

---

## Referee Comment (RC3) · Anonymous Referee #3 · 3 May 2016

General comments :

In this paper, the authors attempted to assess the mass balance of the Quelcaya Ice Cap (QIC) over the last 30 years. From a reduced dataset, they calculated the volume change between 1983 and 2013 by differencing the surface elevations suggesting an average annual mass balance rate of -0.5 m w.e a-1 over this period. Based on calculations of emergence velocities, they claimed that the thinning of the ice cap is due to an increase in melting and sublimation of 1-2 m w.e a-1.

However, the novelty of this paper and the relevance of the results can be questioned:

1) The volume change of QIC has been obtained previously between 1963-1978 and 1978-1991 from photogrammetry (Thompson et al., 2006). In the present paper, the elevation changes are obtained from 46 sites (mainly in the accumulation zone close

to the summit and 5 sites below 5500 m) between 1983 and 2013 on one longitudinal profile only. The added value of this new value can be largely questioned given the poor dataset shown in this study.

2) The new result of this paper could be relative to the cause of this thinning. From emergence velocities calculations, the authors deduced an increase in melting and sublimation of 1-2 m w.e a-1. However, these results are very fragile for the following reasons. First, the emergence velocities determinations and mass balance changes are supported by very few measurements and very large assumptions. 9 stakes measurements are available to assess the emergence velocities (line 2, p. 9) in 1983/1984 in accumulation zone and 3 stakes in ablation zone only. Moreover, in ablation zone, the measurements of ice flow velocities (for emergence velocities calculations) have been done over a duration of one month and extrapolated to an annual rate. Second, the density is unknown in the accumulation zone. The mass balance calculated in accumulation zone from continuity equation depends strongly on density assessment of the firn. The density used for the calculations of emergence velocities contains large uncertainties (Fig. 7). The inferred mass balances are strongly affected by these uncertainties. In this paper, the calculation of uncertainties has not been clearly explained. Third, the emergence velocities (and the inferred mass balances) are also strongly affected by the uncertainties on the slope. In this paper, the authors do not provide any information about the measurements of the slope, the distance on which the slope has been measured. It seems that the uncertainty on the slope has not been taken into account despite the strong impact on the results.

3) This paper suffers from large assumptions and large uncertainties: the calculations of uncertainties are not explained properly. In addition, I believe that all the uncertainties have not been taken into account. From this paper, it is not possible to assess properly the impact of the large assumptions on the uncertainties. What is the impact on slope uncertainties ? The authors did not explain properly the uncertainty calculations of inferred mass balance (bs from Equation 5) taking into account the uncertainty

on ws, us, slope, thickness change and density, and the duration of measurements (one month in ablation zone). The error bars are reported in Figure 6 without any explanations. In addition, regarding the error bars, are the calculated mass balance changes significant? It seems that the uncertainties are the same order of magnitude of the estimated values.

4) This paper suffers from confusion and vagueness: for instance, the results shown in Figure 5 and Figure 6 come from calculations which are not explained properly. It is confusing. From these Figures and from the manuscript, I understood that the emergence velocities calculated in 1983/1984 (blue line) shown in Figure 5 have been calculated from direct measurements w-utg$\alpha$. The mass balance of 1983-1984 shown in Figure 6 (green line) come from these emergence velocities and are similar to emergence velocities shown in Figure 5 (blue line) corrected with density (If it is the case, I do not understand why the zero mass balance is above 5400 m (Fig. 6) while the zero emergence velocity is below 5400 m (Fig. 5)). I understood that the mass balance of 1983-2013 (Fig. 6, red line) have been obtained from the difference between dH1983-2013 and the emergence velocities 1983-1984. The explanations given in section 4.1 and 4.2 are very short and very confusing. The authors should provide clearly the meaning (equation) of each curve.

5) Finally, the mass balance change of 1983-2013 are calculated from emergence velocities obtained in 1983-1984 and the elevation changes obtained between 1983 and 2013. In this way, the authors assume that the emergence velocities are constant over the period 1983-2013 which would mean that the flow lines did not change over this period of 30 years. This assumption cannot be supported by the available data. In numerous studies for which the glaciers have been decreased over the last decades, this assumption is not valid. Unfortunately, the main new result of the present paper is based on this assumption. In section 4.2 (line 7, p. 10), the authors acknowledge that the mass balance and emergence velocities could change together between 1983 and 2013 but this assertion is almost ignored in Discussion, and totally ignored in
Conclusions and Abstract.

6) The discussion about the relationship between rate of margin retreat and ablation (lines 4 – 31 page 12) does not provide significant support to the conclusions. It remains very hypothetical and qualitative.

7) A large part of the discussion relative to accumulation zone (section 5.2, lines 14-30, page 13) should be moved in Data section. In addition, it is not clear how these data have been taken into account in the mass balance calculations inferred from emergence velocities measurements (Fig 6). In this study, it seems that the change of density with depth is not taken into account.

In this study, the authors attempted to use old (1983-1984) and recent data ( 2013 and 2015) to infer surface mass balance change over the period 1983-2013. It is obvious that these measurements have not been carried out for the purpose of the present study. Consequently, the dataset is very poor and many measurements are missing. I believe that the conclusions reported in this study cannot be supported by the available measurements given the very large uncertainties which affect the data. The main conclusion of this paper is that the results can be only solved by measuring new surface velocities across the ice cap as acknowledged in Discussion and Abstract (line 15, p. 14 and line 14, p. 1). Indeed I believe that the expected results about mass balance and the cause of mass balance changes need further measurements. Unfortunately, I do not believe that the results shown in this study are sufficient for publication in The Cryosphere.

Specific comments

Numerous specific comments should be needed to improve the clarity of the manuscript. Some specific comments have been mentioned here, although it is not necessary at this stage.

P.1, line 8 : the authors should avoid the term ' mass balance' given the value has been

obtained from elevation changes on one longitudinal cross section only.

P. 1, l. 10 : provide the uncertainty on the increase of melting

p. 1, l. 18 : m of water ?

p. 2, l. 8 : ' returned frozen to the laboratory ' : this kind of detail is not useful and can be deleted. Other details not directly related to the purpose of the present study can be removed from the manuscript.

p. 3, l. 2 : what do the authors mean by ' highly correlated' ? is there a statistical relationship ?

p. 4, l. 7 : which flow model ?

p. 4, l. 9 : any reference for these previous measurements ?

p. 5, l. 1 : avoid the term ' geodetic mass balance '

p. 6, l. 6-7 : specify the instruments (EDM and theodolite).

p. 6, l. 7 : usually, the accuracy of EDM is given by a constant plus a value which depends on the measured length

p. 6, l. 12 : confusing. In my mind, the reciprocal vertical angles measurements should provide the orthometric height difference given that the observations performed with theodolite are relative to the geoid.

p. 6, l. 16 : the uncertainty relative to N is not mentioned. It is probably high. What are the spatial fluctuations of N in the studied area ? It should strongly affect the accuracy of the elevation changes obtained from classical topography in 1983 and from GPS in 2013.

p. 6, l. 7-28 ; the authors mention the accuracy of the elevation measurements only. They do not mention anything about the horizontal angles and the accuracy of horizontal coordinates. However, the accuracy of horizontal coordinates are crucial given that

the elevation changes accuracy depend also on the XY accuracy.

p. 6, l. 30 : specify the instrument and the method (differential ?). Specify also the duration of the GPS measurements.

p. 7, l. 3 : how is 0.36 m obtained ? what is the uncertainty on N ? How does N change over the ice cap ? It is probably badly known.

p. 7, l. 6 : mention clearly QSP1 and QSP2. Add QSP2 in Fig. 2

p. 7, l. 7-8 : how can the authors obtain an uncertainty of 0.01 and 0.03 m given that the uncertainty of global coordinates is 10 cm (p. 6, l. 31) ? The assessment of uncertainties is not clear.

p. 7, l. 11 : the authors should describe the surface mass balance measurements in this Section

p.7, l. 11 : the authors did not describe the terrestrial measurements of 1978 in this section (instruments, location of measurements, accuracy. . .)

p. 7, l. 13-19 : the authors should explain how the uncertainties have been obtained.

p. 7, l. 27 : ' mass loss' ?

p. 7, l. 29-33 : I do not believe that the authors may obtain a ' geodetic mass balance ' from this very limited dataset.

p.7, l. 29 : why 440 kg/m$^2$ ?

p. 8, l.7 : ' assumed two dimensional flow, i.e no tranverse flow ' : it is not necessary to assume no transverse flow. See Equation 8.65 In Cuffey and Parson (2010). Here, the ice flow velocity has been measured in the direction of the ice flow. The horizontal divergence of the ice flow does not change anything.

p. 8, l. 27 : the uncertainties related to the short duration of measurements (one month) are not explained in the manuscript.
p. 8, l.27 : the uncertainty related to the local slope is not mentioned in the manuscript. In addition, the authors did not describe the method to measure the local slope. Which distance is taken into account to measure the slope ?

p. 9, l. 8 : the calculation of the uncertainty (0.28 m) is not explained

p. 9, l. 13 : I understood that the authors calculated the emergence velocities to infer hypothetical steady state surface mass balances in 1983-1984. However the sentence is confusing.

p. 9, l. 1-14 : the section 4.1 is too short. It should provide more explanations and should provide the calculations of the uncertainties (other parts of the manuscript could be strongly reduced)

p. 10, l. 3-9 : the section 4.2 is very very short. It should provide more explanations and should provide the calculations of the uncertainties. It should also provide clear explanations about the results obtained in Figure 5 and Figure 6.

p. 10, l.6-9 : which data could support that the emergence velocities did not change between 1983 and 2013 ? This assumption is not discussed in the manuscript.

p. 10, 11, and 12 : Discussion : a large part of the discussion is not very helpful. The calculation of ablation from the relationship between ablation and margin retreat is too crude to support the previous results given the uncertainties and the approximations related to this relationship.

p. 13 : Discussion A large part of Section 5.2 should be moved to Data section (accumulation and density measurements).

Figure 3 : the authors should explain the meaning of the lines.

Figure 4 : the two vectors of the ice flow velocities between 5600 and 5550 m are turned upward relatively to the slope although these sites are in the accumulation zone.

Figures 5 and 6 : red dash line of Fig 5: it is not clear how this curve has been obtained.

I understood that it is the difference between the emergence velocity1983-1984 and dH1983-2013 but I am not completely sure given it is not explained clearly. I would suggest to mention clearly the calculations relative to the red and blue lines (and red and green lines in Figure 6) in the manuscript.

Figure 7 : the authors should describe clearly in the manuscript how the red points and lines have been obtained

[Figure]

---

## Editor Comment (EC1) · E. Berthier (Editor) · 24 Jun 2016

Dear authors,

The online discussion of your paper 'tc-2016-40' is now closed. Your manuscript has been thoroughly reviewed by three external referees and I want first to acknowledge them for the time they spent on evaluating comprehensively your study.

All referees raised some important issues on the novelty of your study, its structure and the significance of your results. In particular, some of your results are based on assumptions that are not justified and quantified. Among many other assumptions, the fact that the emergence velocities are assumed to remain constant since 1983-84 (the only year when they were measured) is highly problematic if the authors want to infer changes in mass balance for a thirty year period (1983-2013) during which the ice cap

has been completely out of balance with the climate.

In this context, I ask you to answer point-by-point to the general comments of all three referees first and explain how you would proceed if you were asked to provide a revised manuscript. Then, I will decide whether we will move forward with the review process of this study and if I will consider a revised manuscript.

Best regards,

Etienne Berthier

---

## Author Comment (AC1) · 1 Sep 2016

Dear Editor:

Again thank you for the additional time to respond to the reviewer comments. We authors were spread across several continents this summer.

We feel quite strongly that our geodetic results showing thinning and mass balance rate for the Quelccaya Ice Cap to be both robust, unique and certainly worthy of publication for the largest tropical glacier on Earth.

Two general responses will go a long way towards addressing the reviewers concerns. First, reviewers repeatedly noted the lack of details on the uncertainty calculations. Two of the authors have post-graduate degrees in geodesy and are well versed in geodetic

approaches including propagation of errors thorough a sequence of non-linear functional relationships. All the uncertainties reported in the draft manuscript are rigorous. However, to save space we did not include the details. Clearly, the reviewers found this inadequate. This can be rather easily corrected, though a rough estimate is that this material would occupy $\sim 5$ journal pages. The question is (and this goes back to our original decision to leave out much of the detail) should we include this in the main text or is appropriate to include these details in a well-organized supplemental section? Can you advise us on the best approach?

Second, all reviewers provided several good suggestions on re-organization of the paper. We acknowledge and agree with these multiple suggestions and can re-organize the paper, in terms of distinguishing and separating the methods, results, and discussion sections.

However, we are somewhat baffled by some of the tenor of reviews .

We feel that we read plenty of papers, including in The Cryosphere, that determine a geodetic mass balance from ice surface elevation changes (usually based on comparing a DTM from old maps with LiDAR) and then explain and interpret that as best and as conservatively as possible using whatever additional data and evidence are available. That's pretty much what we did and that's science in the real world.

Reviewers seem somewhat fixated upon lack of and uncertainty of the field data used in the analysis of potential cause(s). We concede that the steady-state assumption / constant emergence velocities since 1983 highly-problematic. We can eliminate the steady-state assumption / emergence velocity assumptions. However, we emphasize these assumption/analysis do not change the fundamental geodetic results (and their relevance).

Thus, that leaves the question of how best to handle explaining the robust pattern of thinning which emerges from the study. We feel we can address most of the reviewer concerns with a re-organzation of the sections and adding more details on the uncertainty calculations. We feel we can sharpen some of the discussion on possible cause(s), although our issues with the timing of annual visits/measurements, density data, and lack of pre-2004 climate measurements will always be limitations.

Finally, we welcome your input and advice on reorganizing the paper.

Our (AUTHOR in red) responses are in the attached document.

Best regards,

Dave Chadwell

Please also note the supplement to this comment:
http://www.the-cryosphere-discuss.net/tc-2016-40/tc-2016-40-AC1-supplement.pdf

**Supplement:**

Title: Thinning of the Quelccaya Ice Cap over the last thirty years Manuscript: The Cryosphere Discuss., doi:10.5194/tc-2016-40, 2016 Authors: C.D. Chadwell, D.R. Hardy, C. Braun, H.H. Brecher, and L.G. Thompson

General Comment:

This manuscript provides detailed information about surveying work conducted on the Quelccaya Ice Cap (QIC), which allows the authors to report on thinning and volume loss along a profile of the western part of the ice body over the last 30 years. The recent field work took place in 2013 and 2015, and was compared to a network established in 1983 – the same year the first deep ice cores were recovered from the QIC. The main contribution of this research is to present up to date numbers about thinning and the volume changes that have taken place since the early field work was completed, with the core of the manuscript describing the data collection and surveying methods.

A continuity relation is also presented, with the purpose of providing information about the physical processes controlling the observed changes.

These findings are a little more tenuous than the surveying results, which is evident in the manner in which they are described by the authors.

A decision should probably be made as to whether these latter results are really necessary and/or if they really add to the understanding of the physical processes controlling the observed retreat.

However, if they are not included there is a question mark over whether the surveying results on their own are enough in their present form to warrant publication.

AUTHORS: Given the paucity of any multi-year glacier thinning data from this region of the Andes, we feel quite strongly these data warrant publication.

To aid in that decision the following comments about the manuscript are offered, which the authors may wish to consider should the paper be considered for publication in The Cryosphere.

Specific comments:

Please note that page number is referred to as (P) and line number is referred to as (L).

1.      Structure: The overall structure of the manuscript could be refined to fall within a more conventional format. There is no clear delineation between methods-results- discussion in the manuscript in its present form. The aim and/or objectives should be more clearly defined at the end of the introduction and it would be valuable to provide information about how the manuscript is structured.

AUTHORS: Agree that this will improve the manuscript's structure. The authors thank Reviewer #1 for this helpful comment. A methods section will be added as well as aims and objectives to end of introduction.

2.      The second section describes the data collection, which is followed by geodetic results (section 3). One could argue that the continuity relation and the methods used to characterize the physical processes in the ablation and accumulation zones (Sections 5.1 and 5.2 in the discussion) could be contained in a discrete methods section. The present discussion (Section 5) is not really a synthesis of all the results. The manuscript would be strengthened if a clearer

methods section was defined, followed by key results and a synthesis of the importance of these in relation to previous and ongoing research.

AUTHORS: Agree that the Discussion section should be a synthesis; this change will be incorporated into the changes in structure. The authors thank Reviewer #1 for this helpful comment.

3.     Abstract: The first part of the abstract is generally well written and provides information about the thinning and volume loss of the "unit-volume" (that needs further clarification in the manuscript – see below). May one ask why the first sentence refers specifically to the challenges of acquiring data from "accumulation zones" of tropical glaciers when the results represent changes in elevation along a transect of QIC?

AUTHORS : Change first sentences in abstract: replace " … within their accumulation zones." To ".. at these high altitude, and often remote locations." Originally we put these words in because almost all other glacier studies are restricted to the ablation zones. Clarifying "unit-volume" should be easy, and given the size of Quelccaya we can make a case that our profile is representative of the entire ice cap.

There is no clear definition of accumulation or ablation zones in the manuscript, and no information is given about the areal extent of each of these zones on QIC.

AUTHORS:  Valid point. I suggest we do this based on our 'best guess' of where the ELA is, from field obs and lots of Landsat images (almost certainly between 5400 and 5500 m), then measure area in PS or GPS software to get zone areas.

The latter part of the abstract, starting with the sentence "increasing air temperatures" and those that follow are not strongly supported by the results contained in the manuscript.
In particular, the statement about increasing air temperature being "a major driver of the observed changes" is not demonstrated by results presented in the manuscript. The sentences that follow are also quite "speculative" and not really supported adequately to be deemed as results. There is no insight as to what "steady-state" refers to and the causes of the surface lowering in the accumulation zone are questionable.

AUTHORS:  In terms of temperature, we can present the observed changes in terms of 2 different Reanalysis datasets (ERA and NCEP/NCAR), which would require an additional figure (despite our already mentioning temperature trends on p. 12). Although we don't have in-situ data prior to 2004, there is a strong association between summit AWS measurements and Reanalysis timeseries. In plots of reanalysis data since 1979, the increase is primarily since ~2000, suggesting that we need to be cautious about referring to linear change and/or steady-state.

4.     Introduction: The introduction is very specific to QIC and does not provide any context for the broader motivation for the research. The first paragraph is very detailed and would almost be better suited as a site description in the methods. If retained, then the area of QIC should be provided here, or elsewhere, as reference to QIC being the "largest tropical glacier" is not really adequate. The failure to provide the area in the manuscript prevents readers from fully appreciating what the "ice unit-volume" represents. As noted above, a clearer aim and some structural information about the manuscript would be of benefit to readers at the end of the introduction.
AUTHORS:  Agree. Easy changes.

5.  Data collection: This is detailed and provides readers with the necessary back- ground about the approach. It might be of interest to reveal to readers what parties were responsible for the early measurements versus those responsible in 2013 and 2015. On P6, L7 the accuracy of the EDM instrument is described, which is the first of many numbers in the manuscript that contain an uncertainty. It would be useful to provide additional information about how this and other uncertainties are calculated. All the uncertainties in the manuscript are given as 2 standard deviations ($2\sigma$) from the value presented, but it is difficult to know in many instances how these are actually calculated. Some effort to unpack these, especially for the surveying results would be useful to readers to help strengthen the findings presented. A table that provides a summary of the field measurements, including the number of stakes installed versus those re-measured and/or surveyed, with the dates when this occurred would help support the text provided to describe these important activities.

AUTHORS:  The geodetic methods and error analysis do exist in much more detail, but  were not included to save space.  Clearly all reviewers found this inadequate.   A section (perhaps in the supplement) will be added to explicitly detail equations used in the calculations and error analysis.  Two of the authors have post-graduate degrees in geodesy and the usual rigor was applied and will be shown.

6.  Geodetic results: As noted above, additional information about how uncertainty is calculated in the elevation change results would be useful to readers. To ensure that the volume changes are more meaningful, the authors should consider more clearly defining what portion of the QIC is represented by the 1983-84 survey network (P7, L23-24). "Using a variety of ice thickness measurements made over the past thirty  years" is not very clear and the "unit-volume of ice along this profile" should be constrained in some way, otherwise the volumetric decreases are not that meaningful. The  uncertainty in the annual mass balance is shown to be $\pm$ 0.1 m – again, how is this calculated and does it take into account, for example, uncertainty associated with applying  specified snow densities?

7.

AUTHORS: This section will be expanded to include the details of the field measurements which include careful analysis of the thickness measurements from gravity, mono-pulse radar and modern ground penetrating radar.  This information exists in detail, but again was summarized to save space.   Again, clearly the reviewers found the lack of this information inadequate.
 Despite the 'noise' in our density measurements, we feel we have enough of them to show that near-surface firn density has likely not changed all that much.

8.  Section 4: The use of the continuity equation to relate surface elevation, mass balance and emergence velocities (P9, L6-12) is justified by the close comparison of  the measurements of surface lowering of the summit elevation and those calculated. How confident are the authors that the similarity of the presented means justify the use of the continuity equation to unravel the key physical processes in the ablation  and accumulation zones? The uncertainty in these estimates is in the same order of  magnitude of the values determined. Are they really that similar and how sensitive are  they to the assumptions necessary to derive them?

AUTHORS:  We felt a complete paper would make an attempt to address the causes of the observed thinning.  In our opinion the logical approach would be to collate all the available glaciological field data for the ice cap and then see if these data reveal the cause.  The end result of the analysis is that the existing field data contain enough uncertainty and /or inter-annual variability to prevent a definitive determination of the cause of the observed thinning. This "null" result – while unsatisfactory – is useful in that shows the survey data as the most

sensitive measure of the glacier mass balance change, as well as directing attention to better care with field measurements in the future.

9.      Discussion: The description of the physical processes occurring in the ablation and accumulation zones (sections 5.1 and 5.2) do not read like a "discussion" section. Both sections are quite data rich and some of the data presented fall outside of the previously defined study period (e.g. Section 5.2 (P13, L15-33 – reference to snow pit sampling starting in 1974). The results presented in Section 5.1 do support the notion that increased ablation can account for the observed thinning and margin retreat, but this section does not provide enough evidence to show what physical processes are responsible. In the abstract, it is stated that it is "melting and sublimation above steady-state" – differentiating between these two is extremely important if air temperature is deemed responsible, which is not done in Section 5.1. The physical processes responsible for surface lowering in the accumulation zone remain poorly constrained using the approach adopted (Section 5.2) – the lack of atmospheric observations or application of a climate model (e.g. energy and mass balance modelling) prevents the authors from adequately distinguishing between the importance of changes in accumulation and densification of snow and ice (firnification rate) – acknowledged by the authors (P14, L3-5).

AUTHORS:  In general we agree that we can't really constrain the physical processes very well. Surely we could learn important information by applying an energy and mass balance model, and do that. However, validation of some results would be difficult, if meltwater is indeed percolating way down into firn (and thus cannot be measured). As with his/her prior comment I think we should discard the notion of steady state, for all pertinent rates may well have changed over the 30 years.

10.     Conclusions: The first paragraph and the results presented in relation to surface lowering and volume loss are of interest. However, the second paragraph (like the abstract), starting with the statement that surface lowering is caused by a combination of "melt, sublimation and ice flux rate" does not provide any new insight – one would expect surface lowering to be a combination of these over most retreating glaciers. It raises the question as to how useful the continuity relation is to this research and if the latter part of the research is that valuable. The approach reads a little bit like a "back of the envelope" exercise – it succeeds in providing a framework to assess the possible causes of thinning and volume loss but ends up failing to fully account for the physical processes responsible. The same applies to the results from the accumulation zone – again, the authors acknowledge that the observations are not "statistically significant". If the latter part of this research is to be retained, it requires further justification as to why it is useful and how it provides a building block to ongoing research. Some explanation as to why climatological data are not included to help resolve some of the uncertainty is almost necessary.

AUTHORS:  Yes, we are cautious. This reviewer's comments are insightful, yet suggest that their experience has primarily been with very accessible glaciers and/or modeling rather than fieldwork. As above, we can provide a record of reanalysis temperature at 500 hPa over the full period of record; these data are 'in hand'.

11.     Final comment: The observational data is of interest to readers given the challenges to extract such valuable information from tropical glaciers – especially the largest one remaining on earth. However, the manuscript falls a bit short in adequately accounting for the physical processes responsible for the observed changes. It succeeds in showing some of the likely mechanisms responsible for the observed changes but is this really enough?

AUTHORS:  Yes we feel it is.  Many papers including those published in The Cryosphere

detect surface change using geodetic techniques and report mass balance without determining the cause(s).  Our geodetic observations are rigorous.

Minor technical suggestions
P1, L1:  small t for tropical glaciers P2, Figure 1:  It would be useful to have an insert that shows where QIC is in relation to the regions that surround it.
AUTHORS:  Yes.
* * *
The authors provide data on the thinning rate and volume loss (per unit area of transect) of the world's largest tropical ice cap, the Quelccaya Ice Cap (QIC). They provide  support from the literature for the values observed thinning rates and modeled mass  balance changes. These data are a significant contribution to our current knowledge  about tropical glacier evolution over the past few decades, in a region where such measurements are limited. Additionally, the authors use the continuity relationship to  evaluate various [climatological and dynamical] causes for the observed thinning at the  QIC. Their conclusions about causes for observed thinning (mainly increased ablation  rates in the ablation zone and a possible combination of density changes and changes  in accumulation in the accumulation zone) are supported by their methods, data, and  applications of previous research. The authors also outline areas where additional research could strength the results of their study and the field's understanding of the  responses to climate change, namely whether ice velocities have changed since the  measurements in the mid-1980s. The paper is well thought out and beneficial to the field, but I  have some broad and specific comments on the content and presentation:
I suggest re-formatting on the Introduction. The Introduction provides a detailed and  chronological list of past work at the ice cap. The driving problem and specific questions of the research, however, are not provided in the Introduction.
AUTHORS:  Good suggestion above. The authors thank Reviewer #2 for this helpful comment.

For the application of the continuity equation to enlighten the causes of the QIC thinning,  there were instances where I was not clear on the assumptions being made. Below, in the Specific Comments section, I enumerate these occurrences.
Additionally, much of the results hinge on whether mass fluxes from the ablation zone to  the accumulation zone have changed, which the authors outline as an area for further  research. Are there end-member cases (on ice velocity) that that the authors could  explore to quantify how dependent their conclusions are on whether mass fluxes have  changed? By adding these end-member scenarios, their conclusions on the causes of  observed thinning would be stronger.
AUTHORS:  Assuming he/she meant to say "from the accumulation zone to the ablation zone", we feel it is best to keep our interpretation hypothetical. With only 4 m of thinning over a broad section of the accumulation zone, mass flux is unlikely to have changed much. If anything, an increase has probably offset some of the observed thinning due increased ablation.

The authors provide a valuable data set on ice thinning and mass loss at the QIC and that will aid future research of low-latitude glaciers. The writing and figures are clear  and logical, and the

paper is well supported by the literature. If the authors address the broad comments (above) and specific comments (below), then I would recommend this manuscript for publication.

Specific Comments:

pg. 1, line 2: should you cite a reference on the QIC being the largest tropical ice mass?

Figure 1/Figure 2: Can these figures be combined? I found the quadrilateral network in Figure 1 seems to be important in Figure 2 but slightly distracting in Figure 1 as a stand-alone figure.

AUTHORS: With addition of a location inset in Fig. 1 with which we agree, we think it would be too much to combine everything.

pg., 2, line 16: Is there a citation for the increasing rate of volume loss?

AUTHORS: Yes, implied within text (Brecher and Thompson, 1993) – although given the lake development at QK, we don't think we should emphasize this at all as it may not be directly related to climatic changes.

pg. 5, lines 1 & 2: I'm not sure what this sentence about how 'typically such geodetic mass balances are based' is trying to say. Are your methods A-typical? By ending the introduction this way, I'm left wondering why this hasn't been done before, rather then excited that this paper will fill in the gap. This specific comment ties back to the first general comment.

AUTHORS: We feel this reviewer should realize how rare such careful geodetic surveys were 30 years ago at remote, high altitude glaciers. We think our wording is fine as is.

pg. 5, line 7: Has the EDM data from 1983 not been published before?

AUTHORS: A unpublished master thesis contains summary of the data, but we worked back from the original log books. This reference will be added.

pg. 7, lines 29 & 30: What are the implications of using a step-wise density profile? From Figure 7, I see that the density profile does affect the modeled results on thinning rate. Has a sensitivity analysis been conducted on the assumptions about the density profile?

AUTHORS: We can add this sensitivity analysis as it is not difficult.

pg. 8, lines 1 & 2: The last two listed possible causes for glacier surface lowering seem to contradict one another. If there is an increased flux out of the accumulation zone, must there also be an increased (instead of decreased) flux into the ablation zone; correct? Also, with tropical glaciers, which have large accumulation area ratios, a decreased flux into the ablation zone should mean that more mass stays in the accumulation zone, which encompasses a greater area, and thus increasing the total height of the glacier. Am I missing something?

AUTHORS: These lines refer to the general situation. We could possibly make the statement more clear by writing "In general, a glacier surface at any point can…"

pg. 8, line 19: Is it appropriate to assume negligible basal motion? The ice core records indicate that the QIC is not a cold-based glacier. Thus basal sliding is permissible. I am not sure if previous work has attempted to constrain the sliding versus deformational velocity of the ice cap. How substantial does basal sliding need to be before this assumption is invalid?

AUTHORS: We will use a more qualified word than "negligible", as we know that even at the thin margin there is basal sliding. Up higher, the gradient is so low that one would expect limited basal motion.

pg. 9, line 14: What information does assuming a steady-state height in 1983-1984 provide? This seems to be a key point of your analysis of the thinning rate measurements. Also, how reasonable is this assumption? The Qori Kalis outlet glacier has retreated throughout the direct observational and photographic record. And from Figure 2, in the region of the survey, the ice retreated even between 1980 and 1984. These observed retreats would lead me to believe that

the ice cap was not in equilibrium in 1983 – 1984 with the climate. Of course, assumptions must be made when using models, but what are the reasons for these assumptions and what are possible implications?

AUTHOR:  Valid points. Not a lot of retreat from 1980 to 84, but the figure does suggest. Again, we should consider a steady-state assumption it is more as a discussion framework and contrary to Reviewer #3 comment below has no effect upon our geodetic results and the mass balance rate calculated.

Figure 5: Is it a coincidence of the data that the emergence velocity is zero at ~ 5400 m in 1983-1984 (blue curve)? Or did you prescribe the curve with that value, since that is roughly the ELA from snow line measurements in the late 1970s? Also, what does a lowering of 75 m in the ELA accounting for the observed thinning mean? I don't see this point discussed in the text.

AUTHORS:  We should add, if maintaining continuity equation. Lowering or increasing of 75 m.

Figure 6: Is it appropriate to think of the red dashed line as the specific balance profile today? And is the difference between the two curves is the mass balance anomaly per elevation? How does the 50 m increase in the ELA compare with values in the literature?

pg. 10, lines 13 & 14: This point should also be made in the introduction and the abstract.

pg. 12, line 11: How do you determine the perturbation in the ablation rate? Are you assuming that in Figure 6, the perturbation in the ablation rate is the difference between the green solid and red dashed curve? At ~ 5400 m a.s.l., that difference looks to be ~ 1 m w.e. per a, but at ~ 5300 m a.s.l. (near the ice margin) that difference looks to be closer to 2 m w.e. per a, which is order unity and reasonable for Equation 6. But in the text or the figure captions a bit more information about what the curves mean would be helpful.

AUTHORS:  We will review this and add new discussion.

pg. 13, lines 6 - 9: These assumptions seem inconsistent with the ice core record and influence of ENSO. If I plot the 2003 South Dome ice core record (plus recent snow pits) (https://www.ncdc.noaa.gov/cdo/f?p=519:1P1_study_id:14174) from thermal year 1976 through thermal year 2009 and plot a linear regression, I find a net summit accumulation rate of change of +0.001 m w.e. per year ) (r-squared value of 0.001 and p-value of 0.81). If I regress from thermal year 1984 through thermal year 2009, the net summit accumulation rate of change is + 0.002 m w.e. per a (r-squared value is 0.001 and p-value is 0.84). If I were to just draw a line from the thermal year 1983 to the thermal year 2009, then I see a net summit accumulation rate decrease of -0.018 m w.e. per a, but this method would not capture a trends in the accumulation. During this period, the 1-sigma value of the net annual summit accumulation is ~ 0.4 mw.e. Even if the starting and ending summit mass accumulation values (that I draw the line between) were several-year averages, connecting dots would only be valid if those averages had similar statistics on ENSO events.

AUTHORS:  Really difficult to know what we can say about changes in accumulation rate. We don't entirely follow this reviewer's points here, with crazy-low r-squares, but clearly there are problems with the published accumulation rate… or at least with its precision. For example, the 2003 ice core accumulation for 1998 is impossible, based on satellite imagery and the well-known impact of El Niño that year over the entire Andean region.

pg. 14, line 26: Is 'effective ablation rate' a perturbation in the ablation rate compared to 1983-1984?
In this paper, the authors attempted to assess the mass balance of the Quelcaya Ice Cap (QIC) over the last 30 years. From a reduced dataset, they calculated the volume change between 1983 and 2013 by differencing the surface elevations suggesting an average annual mass balance rate of -0.5 m w.e a-1 over this period. Based on calculations of emergence velocities, they claimed that the thinning of the ice cap is due to an increase in melting and sublimation of 1-2 m w.e a-1.
However, the novelty of this paper and the relevance of the results can be questioned:
1.     The volume change of QIC has been obtained previously between 1963-1978 and 1978-1991 from photogrammetry (Thompson et al., 2006). In the present paper, the elevation changes are obtained from 46 sites (mainly in the accumulation zone close to the summit and 5 sites below 5500 m) between 1983 and 2013 on one longitudinal profile only. The added value of this new value can be largely questioned given the poor dataset shown in this study.
AUTHORS:  No, the volume of QIC has not been measured before; only the volume of the small outlet glacier Qori Kalis.

2.     The new result of this paper could be relative to the cause of this thinning. From emergence velocities calculations, the authors deduced an increase in melting and sublimation of 1-2 m w.e a-1. However, these results are very fragile for the following reasons. First, the emergence velocities determinations and mass balance changes are supported by very few measurements and very large assumptions. 9 stakes measurements are available to assess the emergence velocities (line 2, p. 9) in 1983/1984 in accumulation zone and 3 stakes in ablation zone only. Moreover, in ablation zone, the measurements of ice flow velocities (for emergence velocities calculations) have been done over a duration of one month and extrapolated to an annual rate. Second, the density is unknown in the accumulation zone. The mass balance calculated in accumulation zone from continuity equation depends strongly on density assessment of the firn. The density used for the calculations of emergence velocities contains large uncertainties (Fig. 7). The inferred mass balances are strongly affected by these uncertainties. In this paper, the calculation of uncertainties has not been clearly explained. Third, the emergence velocities (and the inferred mass balances) are also strongly affected by the uncertainties on the slope. In this paper, the authors do not provide any information about the measurements of the slope, the distance on which the slope has been measured. It seems that the uncertainty on the slope has not been taken into account despite the strong impact on the results.
AUTHORS:  Emergence velocity calculations based on the limited interval (up to 1 year) is a valid issue I think, especially given the influence on ENSO.  However the emergence velocities do not affect the main conclusions of the paper of thinning at the summit and mass balance rate.  As noted previously many papers on mass balance completely ignore emergence velocity and it changes.  Slope was measured and used at each site on the glacier (see comments below).

3.     This paper suffers from large assumptions and large uncertainties: the calculations of uncertainties are not explained properly. In addition, I believe that all the uncertainties have not been taken into account. From this paper, it is not possible to assess properly the impact of the large assumptions on the uncertainties. What is the impact on slope uncertainties? The

authors did not explain properly the uncertainty calculations of inferred mass balance (bs from Equation 5) taking into account the uncertainty on ws, us, slope, thickness change and density, and the duration of measurements (one month in ablation zone). The error bars are reported in Figure 6 without any explanations. In addition, regarding the error bars, are the calculated mass balance changes significant? It seems that the uncertainties are the same order of magnitude of the estimated values.

4.     This paper suffers from confusion and vagueness: for instance, the results shown in Figure 5 and Figure 6 come from calculations which are not explained properly. It is confusing. From these Figures and from the manuscript, I understood that the emergence velocities calculated in 1983/1984 (blue line) shown in Figure 5 have been calculated from direct measurements w-utg$\alpha$. The mass balance of 1983-1984 shown in Figure 6 (green line) come from these emergence velocities and are similar to emergence velocities shown in Figure 5 (blue line) corrected with density (If it is the case, I do not understand why the zero mass balance is above 5400 m (Fig. 6) while the zero emergence velocity is below 5400 m (Fig. 5)). I understood that the mass balance of 1983-2013 (Fig. 6, red line) have been obtained from the difference between dH1983- 2013 and the emergence velocities 1983-1984. The explanations given in section 4.1 and 4.2 are very short and very confusing. The authors should provide clearly the meaning (equation) of each curve.

5.     Finally, the mass balance change of 1983-2013 are calculated from emergence velocities obtained in 1983-1984 and the elevation changes obtained between 1983 and 2013. In this way, the authors assume that the emergence velocities are constant over the period 1983-2013 which would mean that the flow lines did not change over this period of 30 years. This assumption cannot be supported by the available data. In numerous studies for which the glaciers have been decreased over the last decades, this assumption is not valid. Unfortunately, the main new result of the present paper is based on this assumption. In section 4.2 (line 7, p. 10), the authors acknowledge that the mass balance and emergence velocities could change together between 1983 and 2013 but this assertion is almost ignored in Discussion, and totally ignored in Conclusions and Abstract.

AUTHORS: We disagree that our interpretation is so dependent upon emergence velocity calculations, but we can clarify the text.

6.     The discussion about the relationship between rate of margin retreat and ablation (lines 4 – 31 page 12) does not provide significant support to the conclusions. It remains very hypothetical and qualitative.

7.     A large part of the discussion relative to accumulation zone (section 5.2, lines 14-30, page 13) should be moved in Data section. In addition, it is not clear how these data have been taken into account in the mass balance calculations inferred from emergence velocities measurements (Fig 6). In this study, it seems that the change of density with depth is not taken into account.

AUTHORS: As previously responded to above.

In this study, the authors attempted to use old (1983-1984) and recent data ( 2013 and 2015) to infer surface mass balance change over the period 1983-2013. It is obvious that these measurements have not been carried out for the purpose of the present study. Consequently, the dataset is very poor and many measurements are missing. I believe that the conclusions reported in this study cannot be supported by the available measurements given the very large uncertainties which affect the data. The main conclusion of this paper is that the results can be only solved by measuring new surface velocities across the ice cap as acknowledged in Discussion and Abstract (line 15, p. 14 and line 14, p. 1). Indeed I believe that the expected results about mass balance and the cause of mass balance changes need further measurements. Unfortunately, I do not believe that the results shown in this study are

sufficient for publication in The Cryosphere.

AUTHORS:  Unfortunately, we do not believe that this person provided insightful general comments.  Several statements are somewhat erroneous concerning the measurements available at QIC.  As mentioned previously the geodetic results and mass balance estimate are rigorous.  Our attempt to use the continuity equation and available field measurements to determine the cause(s) do not detract from the fundamental results.

Specific comments

Numerous specific comments should be needed to improve the clarity of the manuscript. Some specific comments have been mentioned here, although it is not necessary at this stage.

P.1, line 8 : the authors should avoid the term ' mass balance' given the value has been obtained from elevation changes on one longitudinal cross section only.

P. 1, l. 10 : provide the uncertainty on the increase of melting

p. 1, l. 18 : m of water ?

p. 2, l. 8 : ' returned frozen to the laboratory ' : this kind of detail is not useful and can be deleted. Other details not directly related to the purpose of the present study can be removed from the manuscript.

AUTHORS:  Certain details will be removed.

p. 3, l. 2 :  what do the authors mean by ' highly correlated' ? is there a statistical relationship ?

p. 4, l. 7 : which flow model ?

p. 4, l. 9 : any reference for these previous measurements ?

AUTHORS:  A unpublished master thesis contains some of the data. This reference will be added.

p. 5, l. 1 : avoid the term ' geodetic mass balance '

p. 6, l. 6-7 : specify the instruments (EDM and theodolite).

AUTHORS:  Wild T2 Theodolite,  AGA Geodimeter 112, both were calibrated by the authors.

p. 6, l. 7 :  usually, the accuracy of EDM is given by a constant plus a value which depends on the measured length

AUTHORS:  As mentioned in the text the EDM instrument was calibrated (seven times) at National Geodetic Survey Calibration Baselines.  Thus rather than relying on the manufacturers published error for this model we determined the values for this specific instrument.  We found the bias and scale errors are not significantly different than zero with about 1 mm and 1 ppm uncertainty.  We found a variance estimate for this instrument for distances up to 2 km to be +/- 4.2 mm (2-sigma) with no significant error as a function of length.  Undoubtedly beyond 2 km there is a length dependence, but field measurements on QIC are under 2 km, and most under 1 km.  We confined the calibration over the range of lengths actually measured at QIC.

p. 6, l. 12 : confusing. In my mind, the reciprocal vertical angles measurements should provide the orthometric height difference given that the observations performed with theodolite are relative to the geoid.

AUTHORS:   Deflection of the vertical values are available from the Earth Gravitational Model 2008 (EGM2008).  The values at each specific measurement location was calculated and applied to provide the ellipsoidal height difference.

p. 6, l. 16 : the uncertainty relative to N is not mentioned. It is probably high. What are the spatial fluctuations of N in the studied area ? It should strongly affect the accuracy of the elevation changes obtained from classical topography in 1983 and from GPS in 2013.

AUTHORS:  N, its uncertainty and its variation are all given for EGM2008 in the referenced paper (Pavlis et al., 2012).  Values of N were calculated at each specific measurement location; thus any variation of N in the EGM2008 model is accounted for in the elevations.  The uncertainty in N is at the decimeter level and is included in the final error estimate of the elevation.   It should be noted that because the measurements in '83/84 and 2013/15 refer to physical benchmarks QSP1 and QSP2 errors in the absolute values in N are removed.  Only the variation in N from QSP1/2 to the summit, a distance of ~ 6km is of concern.   Deflection of the vertical uncertainty

is on the order of 1.3 arc-seconds which is the equivalent of ~ 3 cm over 6 km.  Given the smallest observed thinning was 4 m, N and deflection of the vertical errors are well below the elevation changes.

p. 6, l. 7-28 ; the authors mention the accuracy of the elevation measurements only. They do not mention anything about the horizontal angles and the accuracy of horizontal coordinates. However, the accuracy of horizontal coordinates are crucial given that the elevation changes accuracy depend also on the XY accuracy.

AUTHORS:  Horizontal positions error is at the decimeter level for both the terrestrial and GPS based positions.   This is reflected somewhat by velocity error ellipses in Fig.2.    A more thorough documentation of the horizontal errors will be presented either by text, table or plot.

p. 6, l. 30 : specify the instrument and the method (differential ?).  Specify also the duration of the GPS measurements.

AUTHORS: Dual frequency GPS recording both phase and code measurements using a Septentrino receiver.   Processing method is long-range kinematic using either network mode with tracking stations around South America or with Precise Point Positioning (PPP) using NASA JPL GIPSY s/w.   Duration of measurements is approximately 10 minutes at each site.

p. 7, l. 3 : how is 0.36 m obtained ? what is the uncertainty on N ? How does N change over the ice cap ? It is probably badly known.

AUTHORS:  As noted above both N and deflection of vertical are known to a precision well below the smallest measured thinning of 4 m.   We will add a plot of N and deflection of the vertical for the area covered by Fig. 1.

p. 7, l. 6 : mention clearly QSP1 and QSP2. Add QSP2 in Fig. 2

AUTHORS: Will add QSP-2 to Fig. 2

p. 7, l. 7-8 : how can the authors obtain an uncertainty of 0.01 and 0.03 m given that the uncertainty of global coordinates is 10 cm (p. 6, l. 31) ? The assessment of uncertainties is not clear.

AUTHORS:   Standard GPS baseline solution that using the dual frequency phase and code to estimate the positions of QSP-2 relative to QSP-1.   This gives the relative precision between QSP-1 and QSP-2 which is required to make a comparison to the EDM and theodolite measurements between QSP-1 and QSP-2.  Global uncertainty remains 10 cm.   More details will be given on the uncertainty calculations.

p. 7, l. 11 : the authors should describe the surface mass balance measurements in this Section

AUTHORS:  A more detail explanation including the specific points used will be included.

p.7, l. 11 : the authors did not describe the terrestrial measurements of 1978 in this section (instruments, location of measurements, accuracy...)

AUTHORS:   Kern DKM2 one-arc second theodolite angle intersection measurements from QSP-1 and QSP-2.  A formal error propagation was conducted to estimate the uncertainty in elevation and horizontal position at the 1978 sites on QIC.

p. 7, l. 13-19 : the authors should explain how the uncertainties have been obtained.

AUTHORS: Rigorous formal error propagation using the uncertainties of the field measurements and the analytical relationship to derived measurements (elevation, position).  Much more details will be shown.

p. 7, l. 27 : ' mass loss' ?

p. 7, l. 29-33 : I do not believe that the authors may obtain a ' geodetic mass balance ' from this very limited dataset.

p.7, l. 29 : why 440 kg/m$^2$ ?

AUTHORS:  From actual surface pit measurements.

p. 8, l.7 : ' assumed two dimensional flow, i.e no tranverse flow ' : it is not necessary to assume no

transverse flow. See Equation 8.65 In Cuffey and Parson (2010). Here, the ice flow velocity has been measured in the direction of the ice flow. The horizontal divergence of the ice flow does not change anything.

AUTHORS: Agree, results and interpretation unchanged.

p. 8, l. 27 : the uncertainties related to the short duration of measurements (one month) are not explained in the manuscript.

AUTHORS: The length of measurements is included explicitly in the error propagation and the final errors do reflect measurement durations whether one month, one year or 30 years. More details will be given

p. 8, l.27 : the uncertainty related to the local slope is not mentioned in the manuscript. In addition, the authors did not describe the method to measure the local slope. Which distance is taken into account to measure the slope ?

AUTHORS: Local slope was measured at all sites (p.6, l.33) by measuring the surface elevation at four sites surrounding the "pole" location. A plane is fit to these data to determine the local slope and its uncertainity.

p. 9, l. 8 : the calculation of the uncertainty (0.28 m) is not explained

AUTHORS: Will be added. It is the summation of errors in elevation determined in '83 and '84. This involved a rigorous error propagation that included vertical angles, EDM distances, instrument height measurements from QSP-1 to the summit and back.

p. 9, l. 13 : I understood that the authors calculated the emergence velocities to infer hypothetical steady state surface mass balances in 1983-1984. However the sentence is confusing.

p. 9, l. 1-14 : the section 4.1 is too short. It should provide more explanations and should provide the calculations of the uncertainties (other parts of the manuscript could be strongly reduced)

AUTHORS: More details on the uncertainty calculation will be added throughout the paper.

p. 10, l. 3-9 : the section 4.2 is very very short. It should provide more explanations and should provide the calculations of the uncertainties. It should also provide clear explanations about the results obtained in Figure 5 and Figure 6.

AUTHORS: Section 4.2 will be expanded to include more explanation on Fig. 5, and 6.

p. 10, l.6-9 : which data could support that the emergence velocities did not change between 1983 and 2013 ? This assumption is not discussed in the manuscript.

p. 10, 11, and 12 : Discussion : a large part of the discussion is not very helpful. The calculation of ablation from the relationship between ablation and margin retreat is too crude to support the previous results given the uncertainties and the approximations related to this relationship.

p. 13 : Discussion A large part of Section 5.2 should be moved to Data section (accumulation and density measurements).

Figure 3 : the authors should explain the meaning of the lines.

AUTHORS: Lines show trend of elevation change and rate with elevation.

Figure 4 : the two vectors of the ice flow velocities between 5600 and 5550 m are turned upward relatively to the slope although these sites are in the accumulation zone.

AUTHORS: As stated in line #6 of the caption for Figure 4. "…annual surface velocities plotted in the vertical plane are shown without vertical exaggeration…." The glacier surface is plotted with vertical exaggeration. Thus there is no error, but to clarify we will add the surface slope tick mark at each blue triangle to make the interpretation more clear.

Figures 5 and 6 : red dash line of Fig 5: it is not clear how this curve has been obtained. I understood that it is the difference between the emergence velocity1983-1984 and dH1983-2013 but I am not completely sure given it is not explained clearly. I would suggest to mention clearly the calculations relative to the red and blue lines (and red and green lines in Figure 6) in the manuscript.

Figure 7 : the authors should describe clearly in the manuscript how the red points and lines have been obtained.

AUTHORS: While an explanation is given in the text this will be reinforced with explicit formulaic expressions used in Figs. 5, 6, and 7.

---

## Editor Comment (EC2) · E. Berthier (Editor) · 7 Sep 2016

Dear Authors,

With support from the reviewers, I have been considering your answers to their numerous comments/suggestions. Unfortunately, I do not think you provided sufficiently convincing responses to warrant consideration of a revised version of the manuscript. I was expecting a more detail response letter and it is unfortunate that some of the comments (in particular important point #5 and also points #6 and #7 from Referee #3) have not been addressed.

Among the main weaknesses of the paper/responses is the too speculative attribution of the thinning in the ablation and accumulation section of the transect. (i) In the accumulation zone, Referee #2 made an important comment (pg. 13, lines 6 - 9) about

inter-annual variability in accumulation in relationship to El Nino / La Nina events (high inter-annual variability) and the risk of estimating accumulation trends from only a few years of data and the severe dependency of the trend on the start/end point. This point was not really addressed in your responses. (ii) In the ablation zone, Referee #1 and #3 raised a strong contradiction in the TCD paper. In the one hand, you cited several studies describing strongly decreasing ice fluxes toward the ablation zone (on other mountain glaciers, Span & Kuhn, JGR, 2003 is another excellent reading on this topic), equivalent to decreasing emergence velocities and with thus a strong contribution to the observed thinning. But in the other hand, you still state in the abstract that "thinning is likely caused by a 1-2 m w.e. a$-1$ increase in melting and sublimation above steady-state." A conclusion not supported by your data.

Although the reviewers and I acknowledge your efforts to understand the cause of the thinning along this transect, it appears that you simply do not have sufficiently precise data to conclude convincingly.

Your geodetic measurements of elevation change remain solid and will be even more valuable when the complete error analysis you made will be described. The transect-wide mass balance remains more problematic. You did not justify why you used a very low density of 440 kg/m3 to convert the volume change to mass change in the accumulation area (a justification was requested by Referee #3). Such a low density would imply a drastic change in the density vertical profile between 1983 and 2013 that does not appear to have occurred (read among others Huss, TC, 2013 on this topic). You did not say neither how you would demonstrate that the single profile studied here is representative of the rest of the ice cap (your statement in the response letter: "given the size of Quelccaya we can make a case that our profile is representative of the entire ice cap" is not convincing).

I still strongly believe in the scientific values of your elevation change measurements in this remote and rapidly changing region. They will deserve publication in the future but requires a complete rewriting of the paper that goes beyond the scope of what can be

**TCD**

done in the framework of the present submission.

I am sorry for not being more positive. I hope that the reviewer's comments will help you to move forward with a prompt re-submission elsewhere.

Please do not hesitate to contact me in case you have any questions.

Best regards, Etienne Berthier – TC Editor

PS: Be prudent with statements such as "This reviewer's comments are insightful, yet suggest that their experience has primarily been with very accessible glaciers and/or modeling rather than fieldwork." All three referees that worked on your paper have a strong field experience in remote/harsh environment...
* * *